# Deep sequencing of yeast and mouse tRNAs and tRNA fragments using OTTR

**Hans Tobias Gustafsson[1], Lucas Ferguson[2,3], Carolina Galan[1], Tianxiong Yu[4], Heather Upton[3†], Ebru Kaymak[1], Zhiping Weng[4], Kathleen Collins[2], Oliver J Rando[1]\***

[1]Department of Biochemistry and Molecular Biotechnology, University of Massachusetts Medical School, Worcester, United States; [2]Department of Molecular and Cell Biology, University of California, Berkeley, Berkeley, United States; [3]Center for Computational Biology, University of California, Berkeley, Berkeley, United States; [4]Program in Bioinformatics and Integrative Biology, University of Massachusetts Medical School, Worcester, United States

**Abstract** Among the major classes of RNAs in the cell, tRNAs remain the most difficult to characterize via deep sequencing approaches, as tRNA structure and nucleotide modifications can each interfere with cDNA synthesis by commonly used reverse transcriptases (RTs). Here, we benchmark a recently developed RNA cloning protocol, termed Ordered Two-Template Relay (OTTR), to characterize intact tRNAs and tRNA fragments in budding yeast and in mouse tissues. We show that OTTR successfully captures both full-length tRNAs and tRNA fragments in budding yeast and in mouse reproductive tissues without any prior enzymatic treatment, and that tRNA cloning efficiency can be further enhanced via AlkB-mediated demethylation of modified nucleotides. As with other recent tRNA cloning protocols, we find that a subset of nucleotide modifications leave misincorporation signatures in OTTR datasets, enabling their detection without any additional protocol steps. Focusing on tRNA cleavage products, we compare OTTR with several standard small RNA-Seq protocols, finding that OTTR provides the most accurate picture of tRNA fragment levels by comparison to 'ground truth' Northern blots. Applying this protocol to mature mouse spermatozoa, our data dramatically alter our understanding of the small RNA cargo of mature mammalian sperm, revealing a far more complex population of tRNA fragments – including both 5′ and 3′ tRNA halves derived from the majority of tRNAs – than previously appreciated. Taken together, our data confirm the superior performance of OTTR to commercial protocols in analysis of tRNA fragments, and force a reappraisal of potential epigenetic functions of the sperm small RNA payload.

## Editor's evaluation

This important study applies Ordered Two Template Relay sequencing (OTTR-seq) to characterize tRNA and tRNA fragments in yeast and mouse tissues. The authors benchmark OTTR-seq vs several other methods and show OTTR-seq allows unambiguous identification of tRNAs, tRNA fragments, and their modification patterns. Use of OTTR-seq revealed extensive tRNA cargos in mammalian sperms that have been postulated to transmit transgenerational information.

## Introduction

tRNAs represent the physical embodiment of the genetic code and are broadly expressed in all cell types in the body and across a wide range of environmental conditions. Nonetheless, there is increasing evidence that the cellular repertoire of tRNAs differs between different cell types, and

**\*For correspondence:**
Oliver.Rando@umassmed.edu

**Present address:** †Addition Therapeutics, South San Francisco, United States

within a given cell type can be shaped by external factors from proliferation rate (*Gingold et al., 2014*; *Hernandez-Alias et al., 2020*) to metabolite levels (*Laxman et al., 2013*). Mature tRNAs are also cleaved in response to cellular stressors (*Lee and Collins, 2005*; *Lee et al., 2009*; *Thompson et al., 2008*; *Yamasaki et al., 2009*), and the resulting cleavage products – broadly known as tRNA fragments, or tRFs – are increasingly appreciated as potential regulatory molecules in their own right (*Anderson and Ivanov, 2014*; *Keam and Hutvagner, 2015*; *Su et al., 2020*).

In contrast to most other RNA species, characterization of tRNA and tRF levels by deep sequencing has been hampered by technical difficulties in the synthesis of cDNA. tRNAs are subject to a wide range of covalent nucleotide modifications, with some ~15–20% of all tRNA nucleotides thought to be covalently modified to form species ranging from 5-methylcytosine and pseudouridine to more complex modifications like wybutosine or methoxy-carbonyl-methyl-thiouridine (*Phizicky and Hopper, 2010*; *Phizicky and Hopper, 2015*; *Pan, 2018*). Several of these modifications, particularly N1-methylguanosine (m$^1$G), N1-methyladenosine (m$^1$A), N3-methylcytosine (m$^3$C), and N2,N2-dimethylguanosine (m$^2_2$G), are known to interfere with commonly used reverse transcriptases (RTs) and prevent the synthesis of full-length cDNAs. As a result, until recently, systematic analyses of intact tRNA levels typically relied on microarray hybridization (*Dittmar et al., 2006*; *Dittmar et al., 2004*) to avoid a requirement for reverse transcription. With respect to tRFs, although typical deep sequencing efforts do capture some tRNA cleavage products, they are clearly limited to only a subset of the tRFs in a given sample. For instance, a large number of groups have characterized small (18–40 nt) RNAs in mammalian sperm, with most such studies documenting very high levels of 5′ fragments of a small handful of tRNAs (most notably including Gly-GCC, Glu-CTC, and Val-CAC). Yet Northern blots show that 3′ tRNA fragments are also present in these samples (*Sharma et al., 2018*; *Zhang et al., 2018*,) but are not captured by typical commercial library preparation protocols such as Illumina TruSeq.

A number of methods have been developed in the past few years to enable analysis of intact tRNAs by deep sequencing. For instance, to reduce barriers to reverse transcription caused by secondary structures, Hydro-tRNAseq introduced limited hydrolysis of full-length tRNAs to yield short fragments for cloning and sequencing (*Karaca et al., 2014*; *Gogakos et al., 2017*). Alternatively, several early protocols leveraged the highly processive thermostable group II intron reverse transcriptase (TGIRT) to overcome tRNA secondary structures (*Mohr et al., 2013*), along with enzymatic demethylation of m$^1$G, m$^1$A, and m$^3$C by bacterial AlkB in an attempt to avoid premature RT termination (*Zheng et al., 2015*; *Cozen et al., 2015*; *Dai et al., 2017*). Although these first-generation protocols yielded few full-length tRNA sequences, a substantially improved TGIRT-based protocol – mim-tRNAseq – was recently shown to efficiently capture full-length tRNAs with only modest levels of premature RT termination (*Behrens et al., 2021*). Other relatively recently developed tRNA cloning protocols include YAMAT-seq (*Shigematsu et al., 2017*), LOTTE-seq (*Erber et al., 2020*), QuantM-seq (*Pinkard et al., 2020*), nano-tRNAseq (*Lucas et al., 2024*), and LIDAR (*Scacchetti et al., 2024*). We consider advantages and disadvantages of these various methods in the *Discussion*; for instance, many of the methods mentioned above – YAMAT-seq, nano-tRNAseq, LOTTE-seq, and QuantM-seq – rely on adaptor ligation to the 3′ CCA on the tRNA acceptor stem and are thus suitable for intact tRNA cloning but cannot be used for analysis of tRNA fragments.

To capture an accurate profile of small RNAs with well-defined RNA 5′ and 3′ ends, a novel protocol was developed – Ordered Two-Template Relay, or OTTR (*Upton et al., 2021*) – that exploits an engineered version of the *Bombyx mori* R2 retroelement reverse transcriptase (*Bibiłło and Eickbush, 2002*) for sequential template jumping. In a one-step RT reaction, OTTR joins 5′ and 3′ adaptors to a full-length cDNA copy of input RNA. Benchmarking a large number of protocols for bias in the capture of a defined mixture of synthetic small RNAs revealed significantly lower bias for OTTR than for any commercial protocol (*Upton et al., 2021*). Intriguingly, characterization of RNA in tissue culture cell lines using OTTR captured substantial levels of intact tRNAs, suggesting that OTTR could be a promising protocol for tRNA sequencing applications.

Here, we set out to explore the utility of OTTR for analysis of intact tRNAs and tRFs in several biological systems. We successfully sequenced full-length intact tRNAs from budding yeast, and from mouse testis, and confirmed that a number of specific nucleotide modifications induce mismatch signatures in the tRNA sequencing dataset. We next turned to analysis of small RNA populations in three systems: budding yeast overexpressing the RNaseT2 family member RNY1p, mouse cauda epididymis, and mature cauda epididymal sperm. Comparison of OTTR with several commercial

protocols, coupled with gold standard Northern blot validation, confirmed that OTTR more accurately captures tRFs than either NEBNext, Illumina Truseq, or a typical in-house protocol based on adaptor ligation. In the mouse samples, we show that OTTR captures a far greater variety of tRNA cleavage products, including abundant 3′ tRNA fragments, that are invisible to the majority of other protocols examined. Taken together, our data provide an updated view of the mouse sperm small RNA payload, and highlight the utility of OTTR for analysis of tRNAs and tRNA fragments by deep sequencing.

## Results

### Cloning of full-length tRNAs in budding yeast, and mouse testis

We initially sought to compare OTTR with several commercial protocols for analysis of small (18–40 nt) RNA populations in mouse sperm. Our proof-of-concept datasets revealed abundant nucleotide mismatches at presumed sites of tRNA nucleotide modifications (see below), as observed in multiple prior deep sequencing analyses of tRNAs (*Zheng et al., 2015*; *Cozen et al., 2015*; *Dai et al., 2017*; *Behrens et al., 2021*). We therefore set out first to sequence intact tRNAs to empirically characterize effects of nucleotide modifications on deep sequencing libraries, to help guide bioinformatic analyses of tRNA-derived sequences.

We focused on two biological systems. First, given the ease of growing large quantities of *S. cerevisiae* for validation of our sequencing data by Northern blots, along with the extensive characterization of tRNA modifications in this species, we sequenced intact tRNAs from actively growing budding yeast. Second, mouse sperm are thought to carry a payload of small RNAs dominated by 5′ tRNA halves (*Peng et al., 2012*), and an increasing number of studies have implicated mouse sperm RNAs as potential mediators of intergenerational paternal effects (*Sharma, 2019*). However, given that sperm do not carry substantial levels of intact tRNAs (*Sharma et al., 2018*; *Zhang et al., 2018*), we instead first turned to mouse testis samples as a source of intact mammalian tRNAs.

For each sample, we generated total RNA, then either fractionated RNAs over a spin column to enrich for <200 nt RNAs (*Figure 1—figure supplement 1A*), or gel-purified 60–100 nt RNAs. A second size selection step was added following cDNA synthesis to deplete adaptor dimers and enrich for libraries carrying ~60–100 bp inserts. Resulting OTTR libraries were then sequenced to an average of ~10 million reads. Surprisingly, we consistently recovered more full-length tRNAs when using the more lenient RNA sizing by *mir*Vana spin column (*Figure 1A*), suggesting that the process of gel-mediated size selection likely results in tRNA fragmentation. We therefore focused downstream analyses on tRNA-mapping reads in the *mir*Vana-sized OTTR libraries.

Initial mapping of OTTR reads to mature tRNA sequences using standard analytical pipelines was hindered by the high numbers of sequence mismatches resulting from reverse transcription 'errors' at modified nucleotides in tRNAs. We therefore turned to the tRAX analytical pipeline (*Holmes et al., 2022*), a mismatch-tolerant pipeline that accounts for the wide range of post-transcriptional modifications to tRNAs that can complicate typical RNA mapping pipelines. For each tRNA, we calculated the percentage of full-length reads, finding that the majority (~70–90% of reads) of tRNAs were full length in both mouse and yeast samples (*Figure 1B–D*, *Figure 1—figure supplement 1B*).

Comparison to a range of prior tRNA sequencing datasets (*Gogakos et al., 2017*; *Zheng et al., 2015*; *Cozen et al., 2015*; *Behrens et al., 2021*; *Shigematsu et al., 2017*; *Erber et al., 2020*; *Pinkard et al., 2020*; *Scacchetti et al., 2024*; *Watkins et al., 2022*; *Scheepbouwer et al., 2023*) revealed that YAMAT-seq was the most efficient protocol for intact tRNA capture, with reads almost completely derived from full length tRNAs (*Figure 1—figure supplement 2*). That said, YAMAT-seq data were very low complexity, with two tRNAs (Lys-CTT, Glu-GTC) representing 82–86% of all tRNA mapping reads across all three cell lines analyzed (not shown). After YAMAT-seq, OTTR and mim-tRNAseq (*Behrens et al., 2021*) were comparable in terms of capturing full-length tRNAs with ~70–90% of tRNA-mapping reads representing full-length sequences, while the remaining protocols captured various types of partial tRNAs potentially attributable to premature RT termination, internal RT priming, or other forms of tRNA degradation or breakage (*Figure 1—figure supplement 2*). Moreover, we find good overall quantitative agreement between intact tRNA levels measured by OTTR and datasets from comparable samples (e.g. actively growing yeast, and mouse testis), including a dataset obtained using the Nanopore-based nano-tRNAseq (*Lucas et al., 2024*) protocol (*Figure 1—figure supplement 3*). Finally, we explored the ability of these various protocols to capture the known

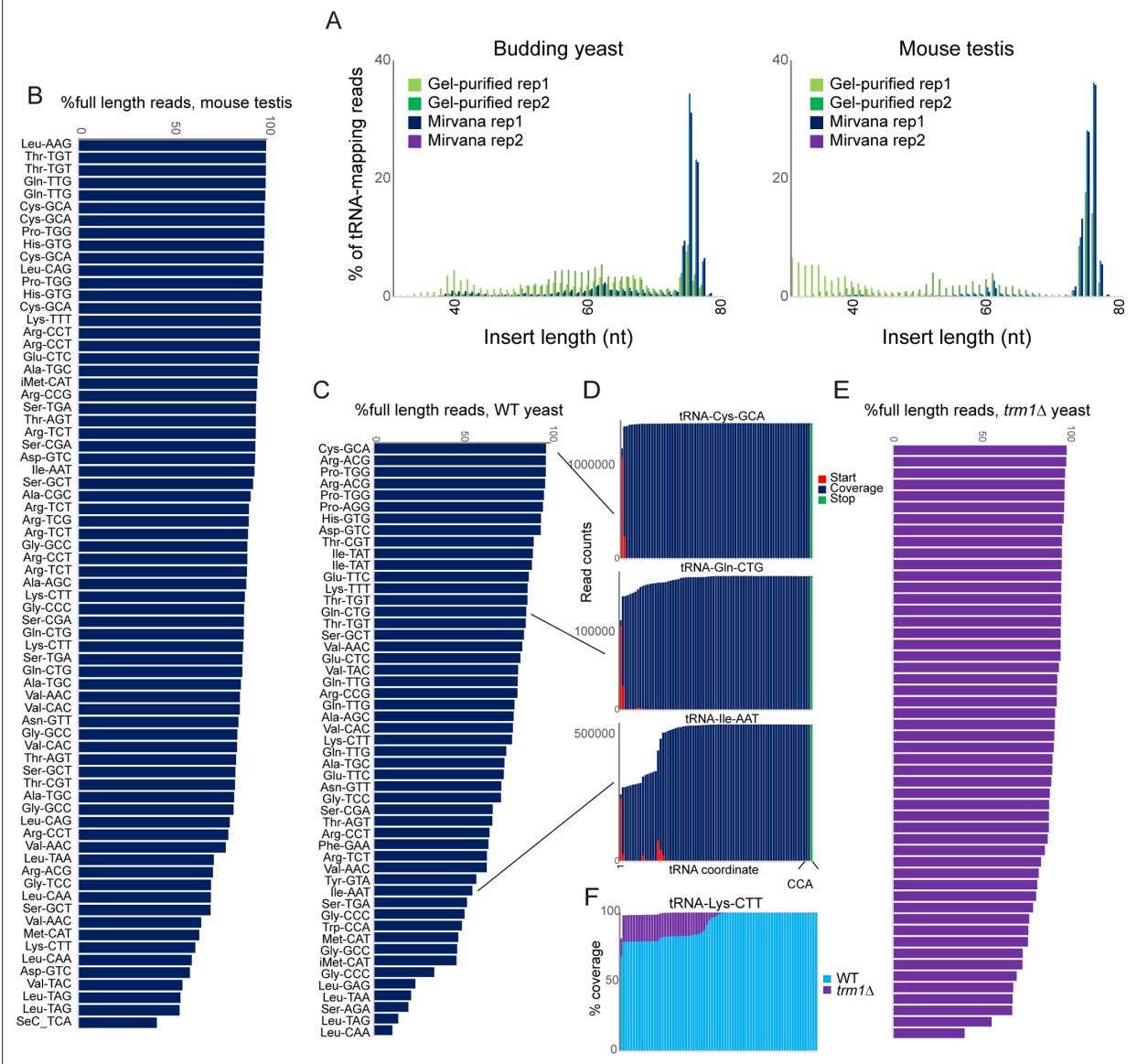

**Figure 1.** OTTR successfully captures full length tRNAs in yeast and mouse. (**A**) Insert length distributions for full-length tRNA OTTR libraries for budding yeast, and mouse testis, as indicated. Libraries were prepared following one of two initial size selection steps: 'Gel' refers to libraries from total RNA subject to acrylamide gel-based purification of 60–100 nt RNAs, 'Mirvana' refers to libraries build using the small (<200 nt) fraction recovered from *mir*Vana RNA spin columns (**Figure 1—figure supplement 1**), and 'rep'1 and 2 refer to replicate datasets. (**B, C**) Efficient capture of full length tRNAs from mouse (**B**) and yeast (**C**) samples using OTTR. For each tRNA species with over 1000 reads, percentage of full length tRNA reads was calculated. See also (**Figure 1—figure supplement 2**). (**D**) Coverage plots for three exemplar tRNAs in the yeast OTTR dataset. Red and green bars show sequence start and stop nt, respectively, while blue bars show sequence coverage internal to a start or stop. WT indicates wild-type. (**E**) Improved full-length tRNA coverage in *trm1Δ* yeast lacking m$^2_2$G, plotted as in panel (**C**). (**F**) Coverage plots for an exemplar tRNA comparing WT and *trm1Δ* yeast. Coverage plots (normalized to the coverage at the tRNA 3' end for each library) are superimposed, with light blue WT over purple *trm1Δ*reads; purple thus highlights the differential between libraries.

The online version of this article includes the following source data and figure supplement(s) for figure 1:

**Figure supplement 1.** Enrichment of short and long RNA populations.

**Figure supplement 1—source data 1.** Source data shows the original gel without any obscuring annotations.

**Figure supplement 2.** Full length tRNA capture across protocols.

**Figure supplement 3.** Quantitative comparison of tRNA levels across methods.

correspondence between tRNA gene copy number and tRNA abundance, with mim-tRNAseq exhibiting the best performance ($R$=0.96), followed by OTTR ($R$=0.5–0.7 across datasets), followed by ARM-seq, Lotte, and nano-tRNAseq ($R$=0.4–0.5 for all three protocols).

Visualization of sequence coverage and read start and stop locations for several tRNAs (*Figure 1D*) suggested that known nucleotide modification sites, including the common $m^1G$ modification found at position 9 of many tRNAs (eg, tRNA-Ile-AAT), could be barriers to reverse transcription. To directly test whether premature termination is affected by nucleotide modifications, we prepared full-length tRNA libraries from *trm1Δ* yeast lacking the methylase responsible for $m^2_2G$ (*Hopper et al., 1982*). We find further gains in the efficiency of full-length tRNA capture in this strain background (*Figure 1E and F*), suggesting that this nucleotide modification presents a partial barrier to reverse transcription (see below).

## Signatures of nucleotide modifications in full-length tRNA sequences

Many of the nucleotide modifications in tRNAs involve chemical alterations that affect the pattern of hydrogen bond donors and acceptors at the base pairing interface. As a result, 'incorrect' nucleotides (relative to those expected from tRNA genomic sequences) can be incorporated into cDNA at these positions during the process of reverse transcription, resulting in a 'mutation/misincorporation' signature for modified nucleotides in deep sequencing data. Examination of individual yeast tRNAs revealed high levels of misincorporation at multiple positions throughout the tRNA (*Figure 2A*), with mismatches localized at various known modification sites, including the expected mismatches at $m^1G$, $m^2_2G$, $m^3C$, and $m^1A$ nucleotides. Examination of the same tRNA species in our *trm1Δ* dataset revealed the expected loss of nucleotide misincorporation at G26 in this mutant (*Figure 2B*), confirming that the $m^2_2G$ nucleotide modification is responsible for the mutational signature at this position.

To explore the effects of modified nucleotides on RT misincorporation globally across all tRNA species and all positions, we plotted 'mutations' for each nucleotide position across all tRNAs in wild type and *trm1Δ* yeast, and in mouse testis (*Figure 2C–E*). Consistent with prior observations (*Behrens et al., 2021*; *Pinkard et al., 2020*), we observed high levels of misincorporation at tRNA positions 9, 26, and 58, corresponding to well-known locations of $m^1G$, $m^2_2G$, and $m^1A$, respectively. Moreover, as seen for individual tRNAs (*Figure 2B*, red arrows), we find that the mutational signature at position 26 is completely lost in the *trm1Δ* background (*Figure 2D*, red arrow), confirming the causal link between the $m^2_2G$ modification and the misincorporation signature at this position. Closer examination of the specific tRNA species exhibiting mismatches at any given position in yeast (*Figure 2F*) confirmed that mismatches were only observed in the subset of tRNAs known to be modified at the position in question (*Dunin-Horkawicz et al., 2006*), as for example alanine tRNAs carry $m^1G$ at position 9 whereas histidine tRNAs do not (*Figure 2F*, leftmost panel).

Beyond the mutational signature observed at known locations of $m^2_2G$, $m^1G$, and $m^1A$, several other locations were associated with sequencing mismatches (*Figure 2C and F*). These included known locations for 3-methylcytidine (32), inosine and 1-methylinosine (34 and 37), and wybutosine (37), as well as lower frequency of misincorporation at several locations currently annotated as unmodified nucleotides in the MODOMICS database (*Dunin-Horkawicz et al., 2006*). Taken together, these data demonstrate the utility of OTTR for analysis of a range of nucleotide modifications.

## Analysis of tRNA cleavage in budding yeast following nuclease overexpression

Turning to analysis of smaller (<40 nt) RNAs, we next set out to benchmark several small RNA cloning protocols in the experimentally tractable budding yeast model system. Cellular tRNAs can be cleaved by a number of different nucleases, including RNase A, T, and L family members, in a variety of species (*Lee and Collins, 2005*; *Thompson et al., 2008*; *Yamasaki et al., 2009*; *Thompson and Parker, 2009*; *Andersen and Collins, 2012*). Conveniently, budding yeast do not encode any RNase A or L family members, and encode a single RNase T2 family member, RNY1p. As RNY1p overexpression has previously been reported to drive high levels of tRNA cleavage (*Thompson and Parker, 2009*), we generated a construct bearing *RNY1* under the control of the galactose-inducible p*GAL1-10* promoter. We confirmed by Northern blot analysis that overexpression of RNY1p lead to high levels of tRNA-Gly-GCC cleavage in our hands (*Figure 3A*, *Figure 3—figure supplement 1A and B*), providing a

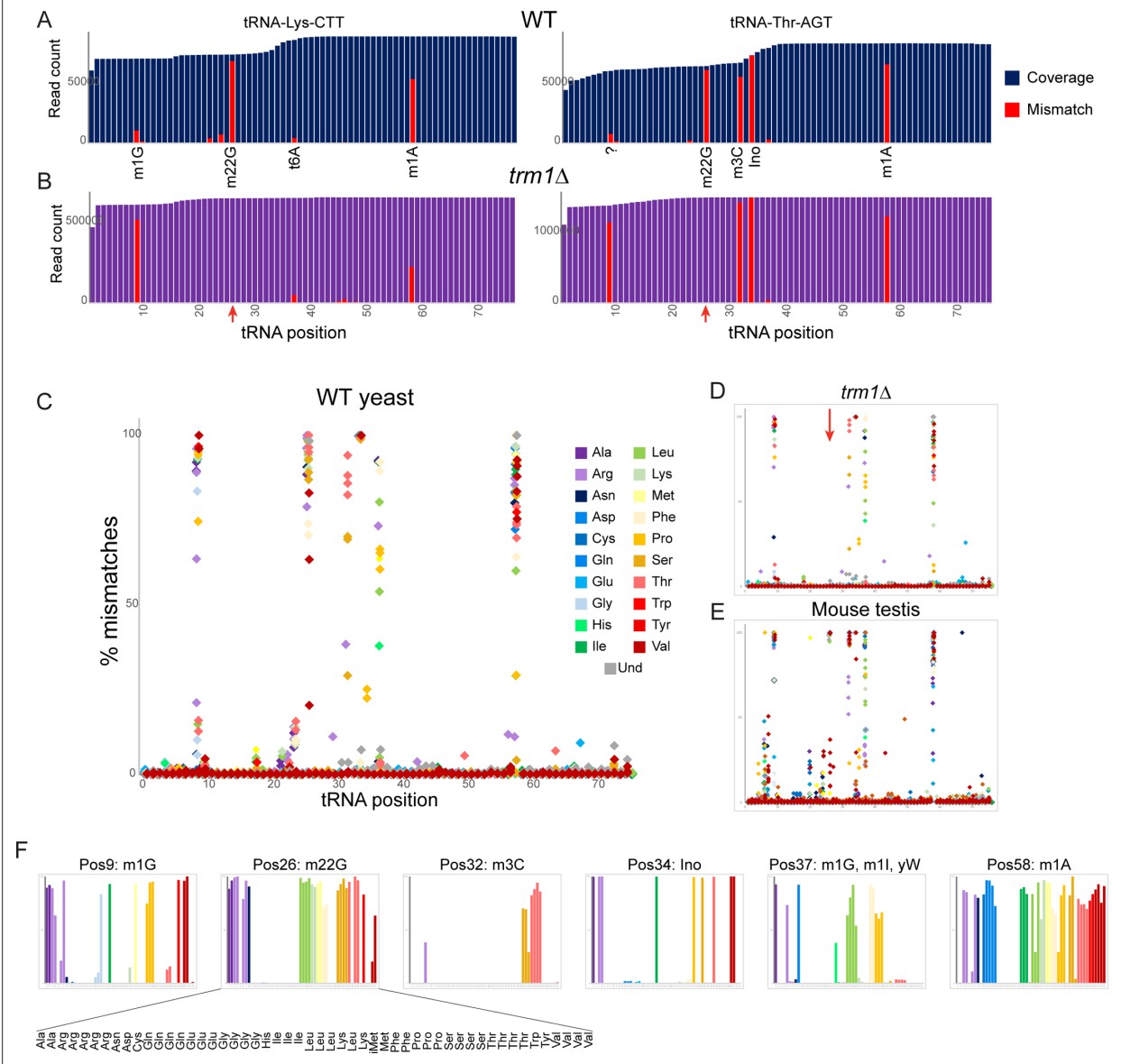

**Figure 2.** Nucleotide modifications revealed by genomic mismatches. (**A**) Sequence coverage of the two indicated tRNAs, with reads matching the genomic tRNA sequence shown in blue, and apparent misincorporations in red. Known nucleotide modifications are shown below each mismatch location. Question mark at position 9 for Thr-AGT indicates no annotated modification at this site in the MODOMICS database (although this site is a common site for the m1G modification in other tRNAs); interestingly, the adjacent nucleotide (position 10) is a known site for N2-methylated guanine in this tRNA. (B) Misincorporation data from the *trm1Δ* dataset for the same two tRNAs as in panel (A). Red arrows indicate loss of mismatches at position 26 in both tRNAs. (**C–E**) Frequency of mismatches across all tRNAs for wild-type yeast (**C**) *trm1Δ* yeast (**D**) and mouse (**E**) OTTR tRNA datasets. In each plot, the % misincorporation is shown for each tRNA position (x axis) for each tRNA species (indicated by colors) with over 2000 reads. Red arrow in panel (**D**) shows the loss of the misincorporation signature at position 26 in *trm1Δ* yeast. (**F**) Detailed view of mismatch frequency at the six indicated nucleotide positions of budding yeast tRNAs. Bar graphs are from the same data as in panel (**C**) The known nucleotide modifications at these positions are annotated.

convenient system for the production of high levels of tRNA fragments for benchmarking small RNA cloning protocols.

To compare small RNA cloning protocols, we overexpressed *RNY1* for six hours in large cultures, purified total RNA, and split the RNA into aliquots for cloning. We generated an initial sequencing dataset to enable comparisons between three basic protocols: (1) a ssRNA ligase strategy (*Fu et al., 2018*) based on RNA ligase-dependent adaptor ligation strategies common to most small RNA-seq protocols; (2) the widely-used NEBNext Small RNA kit; and (3) OTTR. In each case, libraries carrying

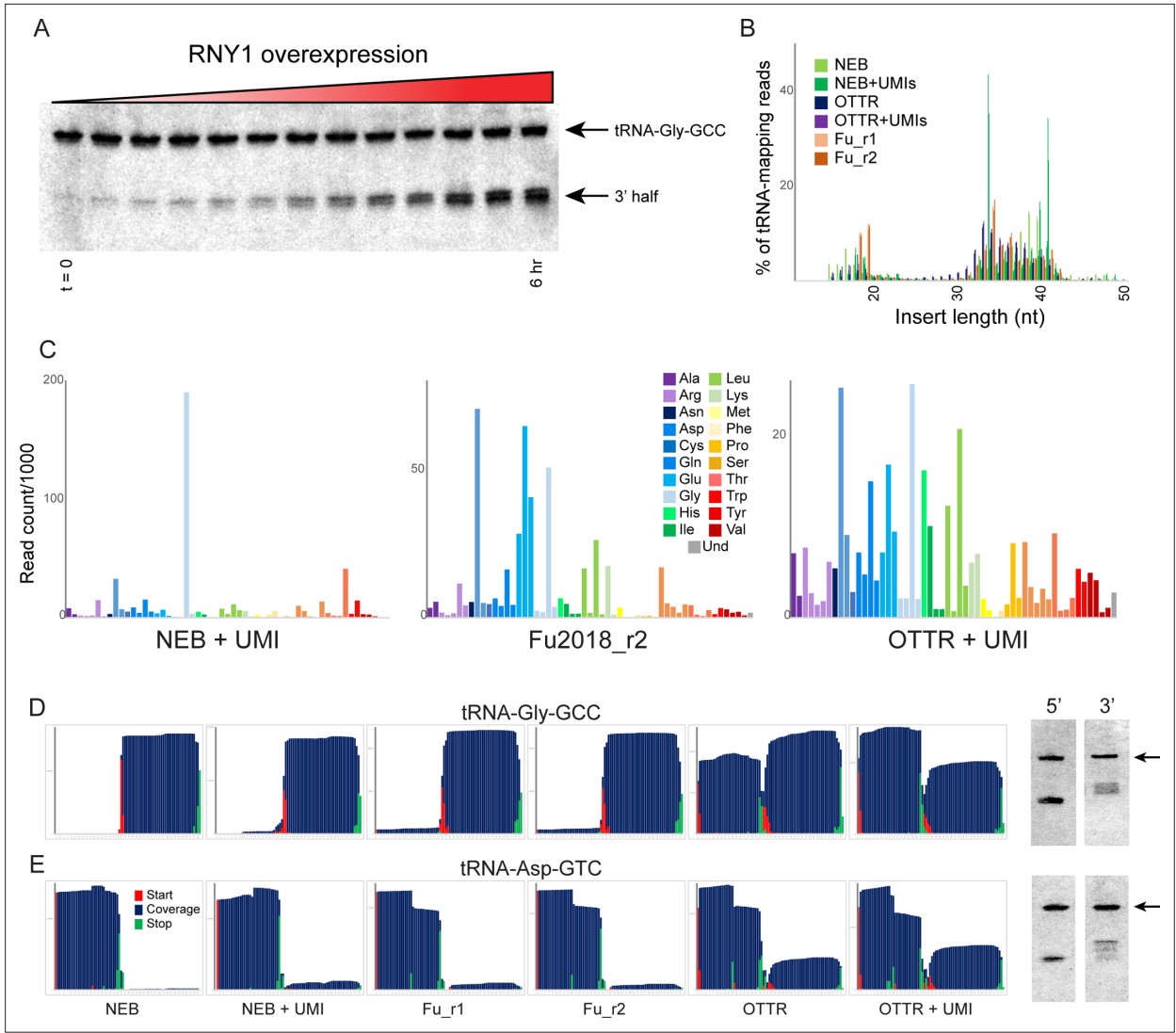

**Figure 3.** Benchmarking OTTR capture of tRNA fragments in budding yeast overexpressing RNY1p. (**A**) Northern blots for tRNA-Gly-GCC 3' end during a time course of RNY1p overexpression (from uninduced to 6 hr induction) in budding yeast. (**B**) Size distributions for tRNA-mapping reads in various small RNA libraries prepared from yeast following six hours of RNY1p overexpression. See also *Figure 3—figure supplement 1*. (**C**) Overall coverage of all tRNA isoacceptors – calculated by summing all reads mapping to a given tRNA species – shown for the indicated small RNA cloning protocols. (**D, E**) Left panels show coverage maps for tRNA-Gly-GCC (**D**) or tRNA-Asp-GTC (**E**) for the six indicated cloning protocols. Each plot shows the distribution of all 5' (start) and 3' (end) ends of the relevant sequencing reads, as well as the cumulative sequencing coverage across the tRNA. Right panels for each tRNA show Northern blots for the 5' side, and the 3' side, of the relevant tRNA (from yeast subject to 6 hr of RNY1p overexpression), as indicated. Black arrow highlights the full-length tRNA band. The deep sequencing datasets from NEBNext and *Fu et al., 2018* protocols capture only the 3' half of tRNA-Gly-GCC, and the 5' half of Asp-GTC, while OTTR captures both 5' and 3' halves. In both cases, Northern blots confirm the validity of the OTTR dataset, with both 5' and 3' halves present at similar abundance for both of these tRNAs.

The online version of this article includes the following source data and figure supplement(s) for figure 3:

**Source data 1.** Original images of Northern blots used in *Figure 3*.

**Source data 2.** Annotated Northern blots for source data.

**Figure supplement 1.** tRF profiling in budding yeast following RNY1p expression.

**Figure supplement 2.** Northern blotting validates OTTR capture of tRNA halves in yeast overexpressing RNY1p.

**Figure supplement 2—source data 1.** Original images of Northern blots used in *Figure 3—figure supplement 2*.

**Figure supplement 2—source data 2.** Annotated Northern blots for source data.

inserts <50 nt were size selected by gel prior to sequencing, to focus on tRNA fragments (*Figure 3B*). All three protocols captured tRFs of similar lengths, albeit with moderate differences between the three protocols – NEBNext was a particular outlier in this regard, with peaks of read counts for specific tRF lengths that were far more prominent in these libraries than in libraries made with the other two protocols.

To compare these cloning protocols in more granular detail, we calculated the representation of all yeast tRNAs in each of the various deep sequencing datasets. As shown in *Figure 3C* and *Figure 3—figure supplement 1C*, we captured relatively few distinct tRNA fragments using the NEB protocol, contrasting with the far wider range of tRFs captured using the Fu2018 protocol and OTTR. Overall, we find that OTTR revealed the greatest diversity of tRFs of all the protocols examined. We extended this analysis by binning tRNA-mapping reads according to the coverage of either the 5′ or 3′ half of each yeast tRNA. This analysis (*Figure 3—figure supplement 1D*) again reveals remarkably few tRNA species efficiently captured by NEBNext, contrasting with somewhat better tRNA capture using the *Fu et al., 2018* protocol, with our OTTR datasets exhibiting the greatest complexity in the range of tRNA fragments captured. The relatively even representation of yeast tRNA species, and 5′ and 3′ halves, is consistent with the expectation that RNase T2 family members like RNY1p should have only modest sequence preferences beyond a preference for pyrimidines present in single-stranded RNA loop regions, and should therefore cleave most tRNAs. In particular, the relatively even distribution of ratios between 5′ and 3′ halves across all tRNAs observed in the OTTR datasets is consistent with Northern blot results (see below), demonstrating roughly similar levels of 5′ and 3′ cleavage products for all four tRNAs assayed.

To enable validation by comparison to an independent measure of tRNA fragment levels, we next examined nucleotide-resolution coverage data for several tRNAs that exhibited substantial differences in measured abundance between the three protocols. *Figure 3D and E* show coverage plots for two exemplar tRNAs – chosen based on dramatic differences in capture of the two halves of the tRNA across the three different protocols – with coverage of the tRNA shown in blue along with the locations of tRNA fragment 5′ and 3′ ends. For example, although all three protocols robustly captured the 3′ half of tRNA-Gly-GCC, OTTR uniquely captured the 5′ fragments of this tRNA that were absent from the other two libraries (*Figure 3D*). To validate these protocols by comparison to an independent ground truth, we assayed the 5′ and 3′ halves of four tRNAs by Northern blotting (*Figure 3D and E*, right panels, and *Figure 3—figure supplement 2*). For the two tRNAs for which the different deep sequencing protocols showed substantial differences in tRNA coverage, OTTR more faithfully captured the tRNA fragment ratio detected by Northern blots. Taken together, our data show that OTTR captures a wider range of tRNA fragments, more even levels of 5′ and 3′ tRNA halves, and better agrees with ground truth Northern blots, all of which strongly support the utility of OTTR for analysis of tRNA-derived small RNAs.

## The small RNA payload of mature mouse spermatozoa

We next turned to analysis of small RNAs in the mouse germline. Scores of studies over the past decade have documented abundant tRNA fragments in mammalian sperm, with the majority of published datasets documenting highly abundant 5′ tRFs, with the 5′ halves of tRNA-Glu-CTC, tRNA-Val-CAC, and tRNA-Gly-GCC representing the three most abundant tRFs in mouse sperm (*Peng et al., 2012*; *Sharma et al., 2016*). However, it has been clear for years that standard deep sequencing analyses are insufficient to fully capture the sperm RNA payload. First, RNA cleavage by RNase A, T, or L family members is known to leave RNA 3′ ends bearing a cyclic 2′–3′ phosphate (which can spontaneously resolve to 2′- or 3′-phosphorylated ends), a modification that prevents RNA ligation during cloning. Indeed, resolving cyclic 2′–3′ phosphates via T4 Polynucleotide Kinase (PNK) treatment (*Honda et al., 2015*) resulted a dramatic shift in captured sperm RNAs (*Sharma et al., 2018*), revealing a far greater abundance and diversity of rRNA cleavage products than previously appreciated, along with longer 5′ tRFs that presumably reflect the primary cleavage site for reproductive tract nucleases (see below). In addition, Northern blotting studies in sperm and epididymis samples revealed the presence of 3′ tRNA halves that are not represented in deep sequencing datasets (*Sharma et al., 2018*; *Zhang et al., 2018*), further emphasizing our incomplete understanding of the mammalian sperm RNA payload.

To directly compare the performance of various small RNA cloning protocols in capturing mouse sperm RNAs, we pooled cauda epididymal sperm from 10 males for total RNA extraction. Total RNAs

were *mir*Vana size selected prior to being split into three large aliquots and either (1) left untreated, (2) treated with PNK in the absence of ATP to catalyze 3′ end dephosphorylation, or (3) treated with PNK and ATP to both resolve 3′ phosphates and to phosphorylate RNA 5′ ends. Each pool was then further split into three aliquots and cloned using either Illumina TruSeq, NEBNext Small RNA, or OTTR. *Figure 4A–C* show insert size distributions, mapping rates to various RNA species, and abundance of various tRNA species, respectively.

Focusing first on commercially available kits, our untreated TruSeq and NEBNext datasets recapitulated features of mouse sperm RNAs documented in many prior TruSeq studies (*Peng et al., 2012*; *Sharma et al., 2016*), including abundant 5′ tRFs deriving primarily from tRNA-Glu-CTC, tRNA-Val-CAC, and tRNA-Gly-GCC (*Figure 4C and D*, *Figure 4—figure supplement 1A*). In contrast, we find that OTTR reveals a far greater range of tRFs than either of the commercial protocols, capturing both 5′ and 3′ tRNA fragments from a much broader representation of tRFs than either of the commercial protocols. These findings are consistent with prior Northern blot studies demonstrating the presence of both 5′ and 3′ tRNA halves in mouse sperm. Moreover, comparing our data to two recent tRF-focused mouse sperm datasets, we find that OTTR captured the most diverse population of tRNA fragments, followed closely by LIDAR (*Scacchetti et al., 2024*), and contrasting with the heavily biased tRF populations captured by PANDORA (*Shi et al., 2021*; *Figure 4—figure supplement 1B*).

## Variations on the OTTR protocol and technical guidance

Finally, given the well-known impact of nucleotide modifications, and 3′ end chemistry (*Sharma et al., 2018*; *Honda et al., 2015*; *Wang et al., 2021*,) on RNA cloning, we set out to characterize the impact of these RNA features on the RNA species captured by the OTTR protocol. As noted above, RNA cleavage by nucleases of the RNase A, T, or L families leaves behind either a cyclic 2′–3′ phosphate or 2′ or 3′ phosphates at the 3′ end of the 5′ fragment. This modification clearly interferes with RNA ligation, but whether it impacts the ligation-independent OTTR protocol is unknown. We therefore first sought to compare the effects of PNK treatment on the spectrum of RNA species captured by various small RNA cloning protocols. Focusing first on the ligation-based cloning methods, we confirm prior reports (*Sharma et al., 2018*; *Shi et al., 2021*; *Wang et al., 2021*) showing that PNK treatment (with or without ATP) resulted in a substantial increase in capture of rRNA-derived fragments (*Figure 4B*), consistent with the hypothesis that rRNA fragments in mammalian sperm are generated by a nuclease of the RNase A, T, or L families. PNK treatment also enabled improved capture of specific cleavage products in the TruSeq and NEB libraries, as assessed by increased overall levels of particular tRFs (*Figure 4—figure supplement 2*).

Compared to the ligation-based small RNA-Seq libraries, the composition of OTTR libraries was far less dependent on PNK treatment (*Figure 4B*). Many of the tRNA fragments that required PNK treatment for capture in TruSeq or NEB libraries were already abundant in untreated OTTR libraries and unaffected by PNK treatment (*Figure 4—figure supplement 2A*). Nonetheless, PNK treatment did lead to improved capture of a subset of tRFs in OTTR libraries (*Figure 4—figure supplement 2A*), and close examination of 3′ cleavage sites revealed capture of longer species for some 5′ tRFs (see for example *Figure 4—figure supplement 2B*, red arrows). Taken together, our findings demonstrate that although OTTR appears to be able to capture small RNAs bearing 3′ phosphates and/or 2′–3′ cyclic phosphate moieties, PNK treatment nonetheless enhances capture of some RNA cleavage products by this protocol and thus should be included for quantitative analyses of tRNA and rRNA cleavage products.

In addition to 3′ end chemistry, a number of common nucleotide modifications in tRNAs are well known to interfere with typically used RTs; indeed, these modifications are generally thought to be a major reason for the historical difficulty of cloning intact tRNAs (*Zheng et al., 2015*; *Cozen et al., 2015*; *Dai et al., 2017*). Although the baseline OTTR protocol successfully captures tRNAs and tRNA fragments without any enzymatic pretreatment, the improved capture of full-length tRNAs in *trm1Δ* yeast (*Figure 1E and F*) suggested that nucleotide modifications might nonetheless impede R2 RT under at least some OTTR library preparation conditions. We therefore characterized the effects of AlkB-mediated nucleotide demethylation (*Zheng et al., 2015*; *Cozen et al., 2015*; *Dai et al., 2017*) on small RNA cloning using OTTR, exploring both intact tRNA and tRNA fragment cloning in both yeast and mouse systems.

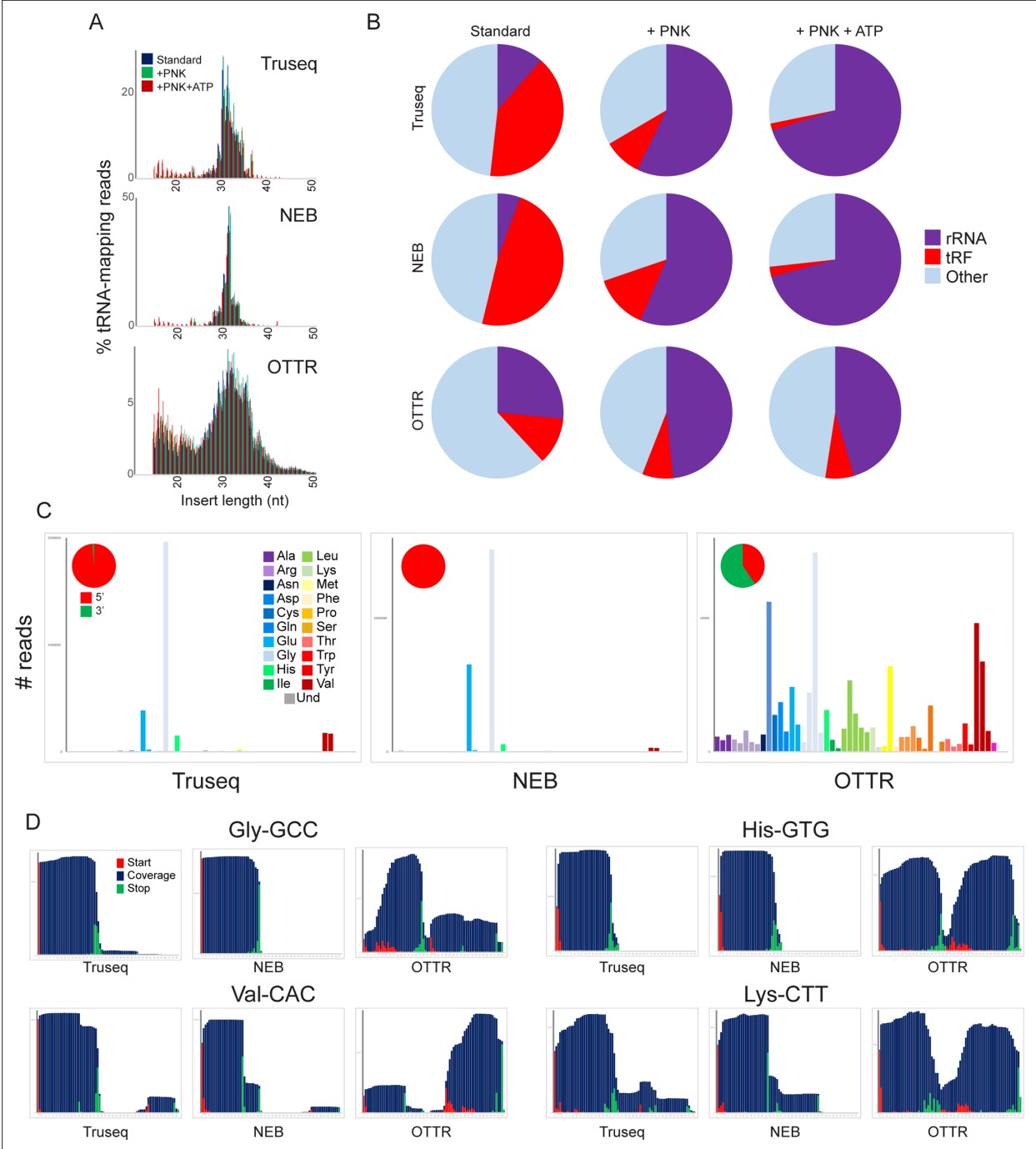

**Figure 4.** A revised view of the mouse sperm small RNA payload. (**A**) Small RNA length distributions, as in *Figure 3B*, for tRNA-mapping reads in various mouse sperm RNA libraries generated using the indicated protocols. (**B**) Pie charts showing overall mapping of each library to the indicated RNA classes. (**C**) Overall coverage of all tRNA species in each dataset, as in *Figure 3C*. Here, each dataset also has a pie chart showing the percentage of tRNA-mapping reads derived from the 5′ or the 3′ half of tRNAs, as indicated. See also *Figure 4—figure supplement 1*. (**D**) Coverage plots for four typical tRNAs, as in *Figure 3D and E*.

The online version of this article includes the following figure supplement(s) for figure 4:

**Figure supplement 1.** Global tRNA coverage in mouse sperm small RNA OTTR libraries.

**Figure supplement 2.** Effects of PNK treatment on tRF levels.

Focusing first on full-length tRNAs, we confirm that AlkB treatment removes $m^1A$ and $m^3C$, as assessed by the near-complete loss of nucleotide mismatches at the relevant positions, but had little to no effect on $m^1G$ or $m^2_2G$ (**Figure 5A**). Overall, we find modest changes in intact tRNA capture following AlkB demethylation, with up to ~threefold altered levels of capture of a variety of tRNAs in the mouse system (**Figure 5B**). Turning next to cloning of tRNA fragments, AlkB treatment resulted in globally increased levels of the majority of 3′ tRNA halves (**Figure 5C and D**), consistent with the $m^1A$ present at position 58 in the majority of tRNAs impeding reverse transcription. Finally, as AlkB had no effect on $m^2_2G$ levels, we also cloned tRNA halves from a *trm1Δ* yeast strain – lacking this modification – overexpressing RNY1p. As with $m^1A$, we find that the $m^2_2G$ modification interferes with reverse transcription by the OTTR RT enzyme, as we observed in our *trm1Δ* samples ~fourfold increased levels of those 5′ tRNA halves that normally carry this modification in wild-type (**Figure 5D**). Taken together, these findings demonstrate that although OTTR can successfully clone modified tRNAs, cloning efficiency is nonetheless affected by $m^1A$ and $m^2_2G$. Although at present there is no enzymatic treatment commercially available to remove $m^2_2G$, AlkB-mediated demethylation of $m^1A$ should be a routine addition to OTTR-based characterization of tRNA levels.

## Discussion

Here, we compared the performance of a variety of small RNA cloning protocols in characterization of tRNAs and tRNA fragments. By several metrics, we find that OTTR outperforms major commercial RNA-seq protocols, and is comparable or superior to a wide range of custom protocols developed specifically to address challenges associated with tRNA capture.

First, considering full-length tRNA cloning, OTTR performs comparably to the mim-tRNAseq method (**Behrens et al., 2021**), which ligates an adaptor to complete or partial RT products, and the YAMAT-seq protocol (**Shigematsu et al., 2017**), which ligates adaptors to mature tRNA acceptor stems before RT (**Behrens et al., 2021**; **Shigematsu et al., 2017**; **Figure 1—figure supplement 2**). Although they use different RTs, all three protocols give some readout of diverse nucleotide modifications including $m^1G$, $m^2_2G$, $m^3C$, $m^1A$, i6A, t6A, inosine, and wybutosine, inferred from mismatches between sequencing read and genomic locus. YAMAT-seq provides the most specific capture of intact tRNAs due to the ligation step involving a Y-shaped adaptor targeting the 3′ CCA of mature tRNAs; subsequent PCR can only amplify full-length tRNAs. However, YAMAT-seq libraries exhibit poor complexity, with the vast majority (82–86%) of tRNA sequences coming from two tRNA species. In contrast, mim-tRNAseq and OTTR exhibit similar levels of intact tRNA capture, similar complexity of species captured, with good ($R$~0.5) correlations between the protocols in measured tRNA levels in yeast (**Figure 1—figure supplement 3A**), albeit with mim-tRNAseq exhibiting better ($R$~0.96) correlation than OTTR ($R$~0.7) to tRNA gene copy numbers. Finally, we note that although LIDAR captures relatively few full-length tRNAs – this is expected from the use of random priming for reverse transcription in this protocol – we also find good correlations between LIDAR and OTTR datasets when counting all partial tRNA reads toward the inferred levels of intact tRNAs (**Figure 1—figure supplement 3B**, top panel).

We additionally show that OTTR more accurately captures tRNA fragments than NEBNext, Illumina TruSeq, an UMI-based custom protocol (**Fu et al., 2018**), and the recently developed PANDORA protocol (**Shi et al., 2021**). Among published mouse sperm datasets, only LIDAR (**Scacchetti et al., 2024**) exhibits similar complexity in tRF capture to OTTR. The OTTR protocol was previously benchmarked using miRXplore – a synthetic miRNA reference standard of 962 small RNA oligos **Upton et al., 2021** – and for ribosome footprints (**Ferguson et al., 2023**). Our benchmarking here extends previous work to tRNAs and tRFs.

We demonstrate here that OTTR is suitable for cloning of both full-length tRNAs as well as tRNA fragments, in contrast to protocols such as YAMAT-seq which require adaptor ligation to the intact tRNA acceptor stem (**Shigematsu et al., 2017**) and are thus limited to full-length tRNA cloning. Similarly, although the Nanopore-based nano-tRNAseq can capture full-length tRNAs, the read numbers in the extant dataset are orders of magnitude lower than obtainable using short read sequencing (~200–400,000 reads in the two datasets in the Supplement from **Lucas et al., 2024**), and the current protocol is not suitable for capturing shorter species like tRNA fragments.

Taken together, the ability to reliably capture a wide variety of RNAs end-to-end, combined with the simplicity of library preparation, highlight the utility of OTTR as a small RNA cloning protocol for

a variety of applications. In our view, OTTR is comparable to mim-tRNAseq (*Behrens et al., 2021*) in applicability: although mim-tRNAseq has not yet been benchmarked against tRNA fragments, the efficient capture of full-length tRNAs, along with the workflow that does not rely on specialized acceptor stem ligation, suggest that this protocol could perform similarly to OTTR for shorter RNAs. LIDAR is also comparable, with advantages and disadvantages relative to OTTR and mim-tRNAseq: the use of random priming for LIDAR results in full-length tRNAs being captured as partial fragments, which makes this protocol a poor choice for applications focused on simultaneous capture of intact and fragmented tRNAs in the same sample. On the other hand, the internal priming enables capture of RNA species with blocked 3′ ends, regardless of the nature of the 3′ block – while 3′ phosphates can be resolved with PNK treatment, other 3′ blocks may be novel or may lack appropriate treatments for resolution, and in these cases LIDAR (and Hydro-tRNAseq *Gogakos et al., 2017*) are uniquely suited to overcoming the blocked 3′ end.

A variety of enzymatic or chemical RNA treatments can be envisioned that would modify the range of RNAs captured by OTTR. We show here that AlkB-mediated demethylation of various tRNA nucleotides – as pioneered in the ARM-Seq and DM-Seq methods (*Zheng et al., 2015*; *Cozen et al., 2015*; *Dai et al., 2017*) – results in significantly improved capture of 3′ tRNA halves (*Figure 5C and D*). Moreover, eliminating $m^2_2G$ via deletion of Trm1 also results in improved capture of relevant tRNA halves (*Figure 5D*). Taken together, for tRNA or tRF quantification, we recommend carrying out OTTR on AlkB-demethylated input RNAs. That said, modifications such as $m^2_2G$ cannot yet be easily removed; it is therefore essential to keep in mind that altered levels of tRNAs or tRNA halves in a given biological system could either reflect changes to tRNA production or stability, or could result from altered levels of inhibitory nucleotide modifications. The latter case should, however, be identifiable by analysis of nucleotide misincorporation signatures (*Figure 2*), at least for those modifications that leave such signatures. OTTR could also be applied to profile tRNA charging, for example by preferential capture of uncharged RNAs (after isolation of total RNA under acidic conditions) or by using aminoacylation as a protection against 3′ nucleotide removal (via periodate oxidation and β-elimination), followed by base treatment to deacylate charged tRNAs. These and other modifications may prove beneficial depending on the goals of any particular study.

## Revisiting the mouse sperm RNA payload

Biologically, our primary interest in benchmarking OTTR was to further explore the still-mysterious small RNA composition of mammalian sperm populations. Over the past decade, scores of studies have largely agreed in defining the mammalian sperm small RNA payload as being dominated by 5′ tRFs, with 5′ ends of tRNA-Glu-CTC, Val-CAC, Val-AAC, Gly-GCC, and Gly-CCC being most abundant (*Peng et al., 2012*; *Sharma, 2019*). rRNA fragments have also been highlighted in several studies of mammalian sperm, although in our experience rRNA fragments proved the most variable between experimentalists, raising the concern that levels of rRNA fragments might be particularly susceptible to artifacts arising during cell lysis, RNA isolation, and/or library preparation.

For several years, it has been clear that the consensus view of mammalian sperm RNAs has been incomplete. We previously showed that removal of 3′ phosphate or 2′–3′ cyclic phosphate modifications revealed a large population of rRNA fragments, as well as slightly longer 5′ tRNA fragments than typically captured, consistent with RNase A or T family cleavage events being responsible for rRNA and tRNA cleavage in the germline (*Sharma et al., 2018*). Moreover, several groups used Northern blots to show that 3′ tRFs are in fact present in sperm, despite their absence from small RNA-seq datasets (*Sharma et al., 2018*; *Zhang et al., 2018*).

Here, we build on these studies, using OTTR to provide the most accurate picture of the sperm small RNA payload to date. We find that sperm carry a population of small RNAs dominated by rRNA fragments, along with both 5′ and 3′ tRNA halves arising from the majority of tRNAs. Smaller populations of microRNAs and piRNAs are also present, consistent with prior reports of the sperm RNA payload. This revised view of the sperm RNA payload raises two major biological questions.

First, our findings undermine the view of sperm RNAs based on the privileged abundance of a handful of specific tRNA 5′ halves, where only specific tRNAs are subject to cleavage, or specific tRNA halves are stabilized and/or selected for trafficking to sperm, thus forming a special population of small RNAs for delivery to the zygote. Instead, our revised view of sperm small RNAs is more consistent with a generalized cleavage of the RNA populations of any typical cell, with the preponderance

of rRNA and tRNA fragments consistent with the abundance of the intact precursor species in developing sperm or typical somatic tissues. Our data do not address the question of whether a given rRNA or tRNA fragment derives from 'in situ' cleavage of rRNAs or tRNAs present at the completion of spermatogenesis – as opposed to their being generated in somatic support cells in the reproductive tract (**Conine and Rando, 2022**) – but our data motivate a reappraisal of the biogenesis of structural RNA fragments in the male germline.

Second, our data raise questions about the biochemical nature of the RNAs delivered to the zygote upon fertilization. Regulatory functions have been identified for multiple solitary 5′ or 3′tRFs in isolation (**Anderson and Ivanov, 2014**; **Su et al., 2020**; **Guzzi et al., 2018**; **Boskovic et al., 2020**; **Kim et al., 2020**; **Pan et al., 2021**), suggesting numerous potential regulatory roles for sperm-delivered tRFs in the early embryo. However, given our finding that for many tRNAs both 5′ and 3′tRNA halves can be found in sperm at similar abundance, it will be important to understand the biochemical context in which these RNAs are delivered to the zygote. Are both 5′ and 3′tRNA halves still associated with one another as part of a nicked tRNA (**Costa et al., 2023**; **Chen and Wolin, 2023**,) or are the two halves dissociated and potentially folded into alternative conformations (**Tosar et al., 2018**), or bound to RNA-binding proteins? Understanding the molecular nature of the tRNA halves that are delivered to the zygote during fertilization has major implications for their potential functions in the early embryo.

Together, our data significantly update our understanding of the sperm epigenome, and motivate re-appraisal of mammalian germline small RNA biogenesis, and of stress and diet effects on sperm RNA populations.

## Materials and methods
### Mouse husbandry and tissue collection
All samples were obtained from male mice of the FVBN/J strain background, consuming control diet Ain-93g, euthanized at 12 weeks of age according to IACUC protocol. For testis samples, both testes were collected from a single FVBN/J male, separated from the epididymis and cleaned of adhering fat, washed with PBS and snap frozen in liquid $N_2$ for later RNA extraction.

For cauda sperm isolation, cauda epididymis samples were collected from 10 males and placed into Donners complete media and tissue was cleared of fat and connective tissue before incisions were made using a 26 G needle while keeping the bulk tissue intact. Tissue was gently squeezed allowing sperm to escape into solution. After incubation at 37 °C for 1 hr, sperm containing media was transferred to a fresh tube and collected by centrifugation at 5000 rpm for 5 min followed by a 1 X PBS wash. To eliminate somatic cell contamination, sperm were subjected to a 1 mL 1% Triton X-100 incubation 37 °C for 15 min with 1500 rpm on Thermomixer and collected by centrifugation at 5000 rpm for 5 min. Somatic cell lysis was followed by a 1 x ddH$_2$O wash and 30 s spin 14,000 rpm to pellet sperm.

### Yeast RNA purification and size selection
The yeast strains used in this study were built on the BY4741 haploid strain background according to standard methods, generating the following strains:

| Yeast Strains | Parent | Genotype | Plasmid |
|---|---|---|---|
| yTG66 | BY4741 | MAT**a** *ura3Δ0 leu2Δ0 his3Δ1 met15Δ0* | pRS416 |
| yTG72 | BY4741 | MAT**a** *ura3Δ0 leu2Δ0 his3Δ1 met15Δ0 rny1Δ::kanMX6* | pTG35 |
| yTG109 | BY4741 | MAT**a** *ura3Δ0 leu2Δ0 his3Δ1 met15Δ0 trm1Δ::kanMX6* | pTG35 |

| Plasmids | Parent | Genotype |
|---|---|---|
| pRS416 | | *URA3* CEN/ARS |
| pTG35 | pRS416 | *URA3* CEN/ARS P$_{GAL1-10}$-*RNY1* |

For all experiments, cells were grown at 30 °C and harvested by centrifugation (2 min at 4000 RPM in 4 °C) and snap frozen in liquid nitrogen.

For full-length tRNA experiments, yTG66 and yTG109 were grown overnight in selective synthetic media containing 2% dextrose, saturated cultures were diluted to OD600=0.1 and grown in selective synthetic media containing 2% dextrose until they reached OD600=0.5–07.

For tRF experiments, yTG72 were grown overnight in selective synthetic media containing 2% raffinose. Saturated cultures were diluted to OD600=0.1 in selective synthetic media containing 2% raffinose and grown until early-midlog (OD600=0.3–0.4). Cells were then centrifuged and diluted in selective synthetic media containing 2% galactose to an OD600 so that they reach OD600=0.5–0.7 in 360 min ('WT' samples were resuspended in selective synthetic media containing 2% galactose, centrifuged, and snap frozen immediately).

Total RNA was prepared by resuspending cell pellets in TNE buffer (50 mM Tris-Cl pH7.4, 100 mM NaCl, 10 mM EDTA) and then vortexed with beads for a total of 2 min (with incubation on ice for a minute after the first minute of vortexing). Equal volume acid phenol chloroform, and SDS to a final volume of 1% was added and samples were vortexed to mix, and then incubated at 65 °C for 7 min, followed by an additional vortex. An additional acid phenol chloroform extraction was performed followed by a chloroform extraction before RNA was precipitated, washed, and resuspended in H$_2$O.

## Sperm RNA purification and small RNA size selection

For mouse sperm RNAs, immediately following cauda sperm purification, sperm RNAs were isolated using the *mir*Vana miRNA Isolation Kit following the enrichment procedure for small RNAs as per manual. Protocol was modified with one half volume of 100% ethanol added to the aqueous phase recovered from organic extraction (recommended volume is one third).

## Northern blots

~3 µg of total RNA from RNY1p-expressing yeast was run on a 15% PAGE-Urea gel at 15 W until dye front reached bottom of gel (~20 min). Probes were as follows:

Arg-CCG 5': TAACCATTGCACTAGAGGAG
Arg-CCG 3': GCTCCTCCCGGGACTCGAAC
Asp-GTC 5': CTGACCATTAAACTATCACG
Asp-GTC 3': CTGACCATTAAACTATCACG
Gly-GCC 5': TACCACTAAACCACTTGCGC
Gly-GCC 3': GCGCAAGCCCGGAATCGAAC
Lys-CTT 5': TACCGATTGCGCCAACAAGG
Lys-CTT 3': GCCCTGTAGGGGGCTCGAAC

## T4 polynucleotide kinase (PNK) treatment

Column purified small RNAs from 10 animals were pooled and split into groups: T4 PNK treatment with ATP, T4 PNK treatment without ATP, and a control no treatment group. T4 PNK treatment with ATP was incubated at 37 °C for 30 min in T4 PNK reaction buffer, 10 mM ATP, and 50U T4 PNK (NEB) M0201S. T4 PNK treatment without ATP was incubated at 37 °C for 30 min in reaction buffer pH6 and 50U T4 PNK (M0201S). All sperm small RNAs samples were then cleaned and concentrated using RNA Clean & Concentrator–5 (Zymo) prior to library preparation.

## AlkB treatment

Up to 5 µg of total RNA was treated with rtStar tRF&tiRNA Pretreatment Kit (Arraystar) according to protocol. AlkB-treated RNA was then cleaned and size selected with RNA Clean & Concentrator (Zymo Research). T4 PNK treatment with ATP was incubated at 37 °C for 30 min. After reaction was stopped, the RNA was cleaned with PCI, precipitated with Isopropanol, and resuspended in 30 µl H2O. AlkB treatment was incubated at 25 °C for 240 min. After quenching the reaction, RNA was cleaned with RNA Clean & Concentrator–5 (Zymo) prior to library preparation.

## Small RNA sequencing

Small RNA sequencing was performed using one of four protocols: TruSeq Small RNA Library Preparation Kit (Illumina), NEBNext Small RNA Library Prep Set for Illumina (NEB), a standard in-house ligation-based cloning protocol (*Fu et al., 2018*; see below), and Collins Lab OTTR Library Preparation Kit (*Upton et al., 2021*). TruSeq and NEBNext library preparation was performed according to manufacturer instructions. In addition, we carried out a slightly altered NEBNext protocol incorporating UMIs to account for potential jackpotting during library preparation. Here, we used the same UMIs as

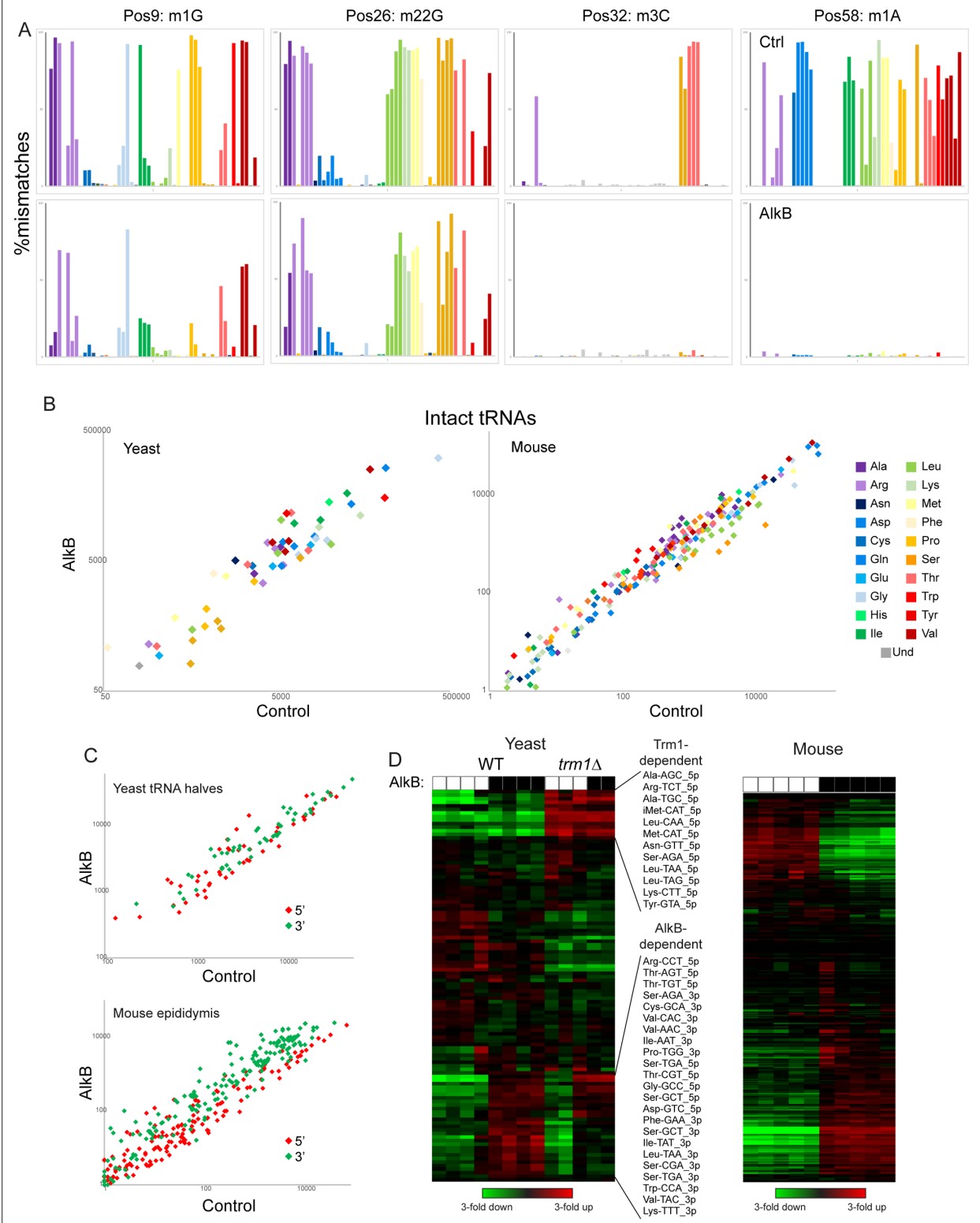

**Figure 5.** Removing nucleotide modifications improves tRNA capture using OTTR. (**A**) Mismatches at the indicated positions are plotted as in *Figure 2F* for full-length tRNA sequencing from yeast. Top row shows control-treated yeast RNAs, bottom row shows AlkB-treated RNAs. (**B**) Scatterplots for intact tRNA abundance in yeast (left) and mouse testis (right), comparing control-treated RNA (x axis) with AlkB-treated RNA (y axis). Some tRNAs, particularly a subset of serine and leucine tRNAs in the mouse sample, exhibit ~threefold differences in abundance following AlkB treatment. (**C**) Scatterplots show abundance of 5' and 3' tRNA fragments in yeast overexpressing RNY1 (top panel), or in the mouse epididymis (bottom panel), before and after AlkB

*Figure 5 continued on next page*

*Figure 5 continued*

treatment. (**D**) Heatmaps showing changes in tRNA fragment representation across the indicated samples for yeast (left panel) and mouse epididymis (right panel).

described in *Fu et al., 2018*, where we incorporated UMIs at both ends (NEB +2 UMIs), as well as 5' end only (NEB +1 UMI).

### *Fu et al., 2018* ligation-based library preparation

A 3' DNA adapter – containing an UMI sequence in 3 nt blocks of random nucleotides separated by pre-defined 3nt consensus sequences, an adenylated 5' end and a dideoxycytosine blocked 3'end – was ligated to size-selected small RNAs using T4 Rnl2tr K227Q (NEB, M0351L) for 16 hr at 25 °C. The ligated product was then purified on a 10% PAGE-Urea gel, followed by gel extraction and ethanol precipitation. The purified ligated product was then ligated to a mix of equimolar 5' RNA adaptors containing UMIs in 3nt blocks of random nucleotides separated by two distinct pre-defined 3nt consensus sequence with T4 RNA ligase (Ambion, AM2141) for 2 hr at 25 °C. The final ligated product was then ethanol precipitated, and cDNA synthesis was performed with AMV reverse transcriptase (NEB, M0277L). cDNA was finally PCR amplified with standard Illumina sequencing primers with Accu-Prime Pfx DNA polymerase for 12–14 cycles. Final PCR product was cleaned up with PCI extraction followed by ethanol precipitation and finally separated by 7.5% PAGE-Urea to remove adaptor dimers. The desired product was excised from the gel and eluted in 750 µl elution buffer overnight at room temperature, followed by isopropanol precipitation and resuspension in 9 µl $H_2O$. Final libraries were pooled and sequenced on Illumina NextSeq 500 with a 75-cycle high-output kit.

### Ordered two-template relay

OTTR was performed as described in *Upton et al., 2021* Briefly, total RNA was size selected either by mirVana (<200 nt) or gel purification (60-100nt). Input RNA was labeled at the 3' end by incubation in terminal transferase buffer containing ddATP for 90 min in 30 °C, followed by an addition of ddGTP and another incubation at 30 °C for 30 min. After heat inactivation of the labelling reaction (65 °C for 5 min), unincorporated ddATP/ddGTP were hydrolyzed by incubation in 5 mM MgCl2 and 0.5 units of shrimp alkaline phosphatase (rSAP) at 37 °C for 15 min. rSAP reaction was stopped by addition of 5 mM EGTA and incubation at 65°C for 5 min. Samples were then incubated in templated cDNA synthesis buffer, adaptors, and dNTPs at 37 °C for 20 min, followed by heat inactivation at 65 °C for 5 min.

cDNA was size selected on an 8% PAGE-Urea gel to minimize adaptor dimer sequencing as described in *Ferguson et al., 2023* Size selected cDNA was PCR amplified for 12–14 cycles with KAPA HiFi hot start (KAPA Biosystems, KK4602). Final PCR product was cleaned up with PCI extraction followed by ethanol precipitation and finally separated by 7.5% PAGE-Urea to remove adaptor dimers. The desired product was excised from the gel and eluted in 750 µl elution buffer overnight at room temperature, followed by isopropanol precipitation and resuspension in 9 µl $H_2O$. Final libraries were pooled and sequenced on Illumina NextSeq 500 with a 75-cycle high-output kit.

UMIs in OTTR are introduced by using an adapter template where the 3'rC is replaced by 3'rC[UMI]. UMIs are sequenced as the first seven bases of read 1, and can be further combined with the +1 Y base of the primer-duplex (*Ferguson et al., 2023*). We used two different versions of UMI adaptors in these experiments. A shorter 5 N UMI, and a longer 12 N UMI.

> 5 N: ACACTCTTTCCCTACACGACGCTCTTCCGATCTNNNNNR/3ddC
> 12 N:ACACTCTTTCCCTACACGACGCTCTTCCGATCTNNNNCGANNNNTACNNNNR/3ddC

During the end stages of manuscript preparation, we observed extensive variability in capture of 5' tRNA fragments from day to day. We ultimately identified the terminal transferase step – the non-templated adenosine addition to input RNA molecules that is the first step of OTTR – as the culprit for failed 5' tRNA fragment capture. We found that robust RNA labeling was compromised by both oxidation of the manganese, which was critical to switch polymerase activity from cDNA synthesis to terminal transferase, and by the oxidation of dithiothreitol (DTT) in the neutral polymerase storage buffer. Manganese oxidation was reduced by incorporating 10 mM sodium acetate pH 5.5 and 28 mM $(NH_4)_2SO_4$, while polymerase storage buffer DTT was replaced with 0.2 mM TCEP, which is known to

be less sensitive to long-time storage at a neutral pH. These optimizations, and other modifications, are unpublished at this time (Ferguson et al, under review) but available from KarnaTeq.com.

## Extended 3′ adaptor ligation

For each of the two ligation-based protocols used for mouse sperm (Truseq, NEB Next), two replicate libraries for mouse sperm were prepared with the additional condition of an 18 hr ligation at 16 °C for the ligation of the 3′ adapter in the attempt to increase ligation efficiency. For OTTR, an 18 hr incubation was added for half of libraries after terminal labeling, during the cDNA synthesis step. However, these interventions had minimal effect on sperm small RNA profiles and both replicates are interchangeable.

## Data analysis

To analyze small RNA sequencing data, we first removed adapters using cutadapt (version 2.9) and PCR duplicate were removed with seqkit (version 0.14.0). The trimmed and deduplicated reads were then analyzed using both an in-house pipeline and the unpublished tool tRAX (version 1.0.0; http://trna.ucsc.edu/tRAX/; *Holmes et al., 2022*) and the results were compared. Firstly, we used Bowtie (version 1.1.0) to map the reads to the annotated rRNA, snoRNA, snRNA, and tRNA sequences in the corresponding species (yeast, mouse, and human) in descending priority, and then the unmappable reads to the respective genomes. The Bowtie parameters used for rRNA, snoRNA, snRNA and genome alignment were '-v 0 k 1'; while the Bowtie parameters used for tRNA alignment were '-y -k 100 −−best −−strata' considering tRNA nucleotide modifications. The abundance of each type of small RNAs was normalized by the total sequencing depth, that is, the total number of small RNA and genome mapping reads in a sequencing library. Secondly, we used tRAX *Holmes et al., 2022* to process the trimmed and deduplicated reads with default parameters; the results largely agreed with what was obtained with our in-house pipeline.

## Comparison to published datasets

Datasets were downloaded and trimmed as follows. If nothing else is noted, adaptor timing was performed with cutadapt 4.1; only reads with a minimum read length of 15 nt and quality scores over 20 were kept for downstream analysis. Trimming details follow for specific datasets.

> ALL-tRNA-seq: (GEO# GSE186736): AGATCGGAAGAGCACACGTCTGAA was trimmed from the 3′-end of the read, followed by the removal of the 4 N UMI located on each side of the insert.
> ARM-seq (GEO# SRP056032): AGATCGGAAGAGCACACGTCTGAA was trimmed from the 3′-end of each read.
> DM-tRNA-seq (GEO# GSE66550): Reads were already trimmed upon download.
> HYDRO-seq: (GEO# GSE95683): TCGTATGCCGTCTTCTGCTTG was trimmed from the 3′-end of the read, followed by the removal of 5nt from the 3′-end.
> LIDAR-seq: (GEO# GSE233343): Trimmed according to instructions on GitHub https://github.com/bonasio-lab/LIDAR (*Shields, 2024*).
> LOTTE-seq: (GEO# PRJNA541863): CGACACTGTCGGTAC was trimmed from the 3′-end of the read.
> mim-tRNA-seq: (GEO# GSE152621): Reads were already trimmed upon download.
> MSR-seq: (GEO# GSE198441) Paired-end reads were merged using Pear/0.9.11. AGATCGGAAGAGCACACGTCTGC was then trimmed from the 3′-end of the read, followed by the removal of UMI sequences on each side of the insert.
> PANDORA-seq: (GEO# GSE144666): Reads were already trimmed upon download
> QuantM-seq: (GEO# GSE141436): TCCAACTGGATACTGGN was trimmed from the 5′-end of the reads, GTATCCAGTTGGAATT was trimmed from the 3′-end of the reads.
> YAMAT-seq: (GEO# SRP096584): ACTGGATACTGG was trimmed from the 5′-end of the reads, GTATCCAGTTGGAATT was trimmed from the 3′-end of the reads.

Each sample was then mapped to the matching reference genome (mm10, hg38, or sacCer3) using tRAX as described above. tRAX output was used to calculate fraction of tRNA-mapping reads representing full-length, 5′ fragments, 3′ fragments, or other (internal fragments and leader/trailer

sequences; *Figure 1—figure supplement 2*). For comparisons of intact tRNA levels between datasets, all reads mapping to a given tRNA were used in the top panels of *Figure 1—figure supplement 3*. (e.g. truncated 5' and 3' fragments were assumed to report on the intact tRNA from which they derived), as the majority of protocols captured relatively low levels (<25%) of intact tRNAs.

## Acknowledgements

We thank P Zamore and I Gainetdinov for the generous gift of primers and technical assistance with the Fu et al 2018 ligation-based RNA cloning protocol, and R Flynn for critical reading of the manuscript and insightful discussions. This work was funded by NIH R01HD099816 (HTG, OJR), F31HD097928 (CG), and the UC Berkeley Bakar Fellows program (KC).

## Additional information

### Competing interests

Lucas Ferguson, Kathleen Collins: is an inventor on published patent applications filed by University of California describing OTTR technology, all of which are also described in peer-reviewed journal publications. Has equity in the company that licensed the OTTR technology (Karnateq). The other authors declare that no competing interests exist.

### Funding

| Funder | Grant reference number | Author |
| --- | --- | --- |
| National Institutes of Health | R01HD099816 | Hans Tobias Gustafsson<br>Lucas Ferguson<br>Carolina Galan<br>Tianxiong Yu<br>Heather Upton<br>Ebru Kaymak<br>Zhiping Weng<br>Kathleen Collins<br>Oliver J Rando |
| National Institutes of Health | F31HD097928 | Carolina Galan |
| UC Berkeley | Bakar Fellows program | Kathleen Collins |

The funders had no role in study design, data collection and interpretation, or the decision to submit the work for publication.

### Author contributions

Hans Tobias Gustafsson, Data curation, Formal analysis, Validation, Investigation, Visualization, Methodology, Writing - original draft, Writing - review and editing; Lucas Ferguson, Resources; Carolina Galan, Conceptualization, Data curation, Formal analysis, Validation, Investigation, Visualization, Methodology; Tianxiong Yu, Software, Formal analysis; Heather Upton, Resources, Investigation; Ebru Kaymak, Investigation; Zhiping Weng, Formal analysis; Kathleen Collins, Resources, Supervision; Oliver J Rando, Conceptualization, Resources, Data curation, Formal analysis, Supervision, Funding acquisition, Visualization, Writing - original draft, Project administration, Writing - review and editing

### Author ORCIDs

Kathleen Collins (iD) https://orcid.org/0000-0003-3172-7088
Oliver J Rando (iD) https://orcid.org/0000-0003-1516-9397

### Ethics

Animal husbandry and experimentation was reviewed, approved, and monitored under the University of Massachusetts Medical School Institutional Animal Care and Use Committee (Protocol ID: A-1833-18).

Decision letter and Author response
Decision letter https://doi.org/10.7554/eLife.77616.sa1
Author response https://doi.org/10.7554/eLife.77616.sa2

## Additional files

### Supplementary files
MDAR checklist

### Data availability
Deep sequencing data are available at GEO, accession #GSE197651.

The following dataset was generated:

| Author(s) | Year | Dataset title | Dataset URL | Database and Identifier |
|---|---|---|---|---|
| Gustafsson HT, Galan C, Yu T, Upton HE, Ferguson L, Kaymak E, Weng Z, Collins K, Rndo OJ | 2022 | Deep sequencing of yeast and mouse tRNAs and tRNA fragments using OTTR | https://www.ncbi.nlm.nih.gov/geo/query/acc.cgi?acc=GSE197651 | NCBI Gene Expression Omnibus, GSE197651 |

The following previously published dataset was used:

| Author(s) | Year | Dataset title | Dataset URL | Database and Identifier |
|---|---|---|---|---|
| Behrens A, Rodschinka G, Nedialkova DD | 2021 | High-resolution quantitative profiling of tRNA abundance and modification status in eukaryotes by mim-tRNAseq | https://www.ncbi.nlm.nih.gov/geo/query/acc.cgi?acc=GSE152621 | NCBI Gene Expression Omnibus, GSE152621 |

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
