## [Editor Report]

This important study applies Ordered Two Template Relay sequencing (OTTR-seq) to characterize tRNA and tRNA fragments in yeast and mouse tissues. The authors benchmark OTTR-seq vs several other methods and show OTTR-seq allows unambiguous identification of tRNAs, tRNA fragments, and their modification patterns. Use of OTTR-seq revealed extensive tRNA cargos in mammalian sperms that have been postulated to transmit transgenerational information.

---

## [Decision Letter]

**Decision letter after peer review:**

Thank you for submitting your article "Deep sequencing of yeast and mouse tRNAs and tRNA fragments using OTTR" for consideration by *eLife*. Your article has been reviewed by 2 peer reviewers, and the evaluation has been overseen by a Reviewing Editor and James Manley as the Senior Editor. The following individual involved in review of your submission have agreed to reveal their identity: Todd M Lowe (Reviewer #2).

Essential Revisions:

1) Better comparison of OTTR to existing methods, such as YAMAT-seq and Mim-tRNA seq. Primary data of Is the measurement of full length tRNA quantitative? The authors observe that the fraction of full length tRNAs measured compared to RT stops is high, like in YAMAT-seq and mim-tRNA-seq. One possibility that the authors detect a high fraction of full length tRNAs is because cDNA products resulting from RT stops are systematically not detected, perhaps due to template switching being prevented by the cDNA template still annealed to the tRNA. YAMAT-seq is notorious in overemphasizing full-length tRNA as only the full-length cDNA product of tRNA is PCR-amplified and sequenced. Mim-tRNA-seq on the other hand, showed that the entire cDNA products BEFORE PCR amplification are indeed mostly full-length. The authors need to show primary data of the cDNA libraries before PCR amplification to justify this claim. In addition, the authors should show without PCR amplification that the OTTR RT is indeed superior in reading through e.g. yeast tRNAPhe yW37 modification (reported in 1C) compared to TGIRT and the thermostable superscript IV used in other recent tRNA-seq protocols.

2) Demonstrate better that OTTR actually distinguishes between 3' tRNA fragment and RT stops. Sequencing reads mapping only to the 3' half are interpreted as fragments.

This is confusing because the method implied that only full length products can be amplified in libraries. The cause and confirmation of the 3' fragments need to be better substantiated.

*Reviewer #1 (Recommendations for the authors):*

Gustafsson et al. describe an application of the recently developed OTTR library prep method to tRNA and tRNA fragments. They present characterization of tRNA-sequencing results from yeast and mouse testis/spermatozoa. The authors claim a significant improvement in the detection of full length tRNAs compared to other techniques; importantly, the authors further focus on the detection of tRNA fragments which are biologically important, but systematically under-detected by existing techniques. The authors show that OTTR detects select tRNA fragments better than a couple of commercial kits. This reviewer shares the author's enthusiasm about the importance of detecting tRNA fragments, especially 3' tRNA fragments, and further share their enthusiasm about the potential for OTTR. However, it is unclear that using OTTR towards the study of tRNA and tRNA fragments is a substantive advance over the several protocols published in the last 3 years. If the authors can more substantively demonstrate how OTTR compares to other methods quantitatively and include additional controls for sequencing results, this could be a method of great utility.

1. Figure 1: Is the measurement of full length tRNA quantitative? The authors observe that the fraction of full length tRNAs measured compared to RT stops is high, like in YAMAT-seq and mim-tRNA-seq. One possibility that the authors detect a high fraction of full length tRNAs is because cDNA products resulting from RT stops are systematically not detected, perhaps due to template switching being prevented by the cDNA template still annealed to the tRNA. YAMAT-seq is notorious in overemphasizing full-length tRNA as only the full-length cDNA product of tRNA is PCR-amplified and sequenced. Mim-tRNA-seq on the other hand, showed that the entire cDNA products BEFORE PCR amplification are indeed mostly full-length. The authors need to show primary data of the cDNA libraries before PCR amplification to justify this claim. In addition, the authors should show without PCR amplification that the OTTR RT is indeed superior in reading through e.g. yeast tRNAPhe yW37 modification (reported in 1C) compared to TGIRT and the thermostable superscript IV used in other recent tRNA-seq protocols.

2. Figure 1: The relative quantification of individual tRNAs needs to be done for reporting any new tRNA-seq method. At the least the authors should compare the yeast OTTR results with that from the mim-tRNA-seq paper.

3. Figure 2: 2A and B show that Lys-CTT has starts near the anticodon loop at t6A, indicating fragments. But this effect is eliminated by trm1 deletion. An alternative from 5' fragments is that the RT reinitiates downstream of a stop induced by a m22G or t6A modification.

4. RNA and read lengths appear inconsistent. First, from figure 1A, is miRvana the best source for capturing small tRNA fragments? More 30-nt material is retained with gel purification. Further, the insert sizes between figures 1A, 3B and 4A are confusing. 1A suggests majority full length tRNA, while figures 3B and 4A crop out this size of read, preventing comparison of relative levels of fragment and full length tRNA. Additionally, figure 1C suggests that Serine and leucine tRNAs are not captured faithful as full length. This could be due to these tRNAs being type II (~90nt long) but sequenced with only 75bp Illumina reads.

5. Can OTTR actually distinguish between 3' tRNA fragment and RT stops? Figure 1B is the strongest evidence to distinguish, and my reading is that reads mapping only to the 3' half must be fragments since this method doesn't suffer from RT stops. This reasoning is insufficient to interpret 3' fragments in unknown biological systems, which severely limits the utility of this exciting approach.

6. The biological advances of this work are unclear. The primary biological discovery presented is that the suite of small RNAs in mouse testis/sperm is different from previous measurements. Even for this result no Northern blot validation is provided for the OTTR data.

*Reviewer #2 (Recommendations for the authors):*

The experimental design for this study was effective in choice of samples and treatments compared. Use of both yeast and mammalian samples that had been previously analyzed was a wise choice by which to observe tRNA pool complexity and misincorporations caused by modifications. We also appreciate the use of tRAX software, which specifically deals with the complexity of misincorporations caused by tRNA modifications, tRNA mapping ambiguity, and differential expression (full disclosure: developed in the Lowe lab). The use of trm1 mutants, inducible RNY1 overexpression, and PNK treatment were all effective experimental contexts showing the relative performance of the various sequencing methods tested. Some readers may feel the study was a bit limited in scope in that it did not do direct sample comparisons with the other leading methods for analyzing full length tRNAs (DM-tRNA-seq, mim-tRNA-seq and/or YAMAT-seq), relying instead on data previously published by other groups. However, in the authors' prior studies, they focused heavily on tRNA fragments in the mouse germline and utilized traditional small RNA-seq methods like TruSeq, which yielded very high-profile results based on high abundance of 5' halves of only a handful of tRNAs. This new study successfully argues the importance of re-examining those prior results given the effectiveness of this new generation of sequencing methods. OTTR-seq is arguably at the front of these newer methods, given its flexibility to measure both tRNAs and tRNA-derived small RNAs at the same time, both in terms of abundance and the misincorporations caused by Watson-Crick face interfering modifications. If all future tRNA-seq studies used OTTR-seq (or equally well performing methods), the tRNA field would take a major step forward.

Aside from easily addressed issues with figures, the only serious confusion that needs to be addressed is the assessment or interpretation of "early termination" sequencing reads. An analysis of premature termination was shown in Figure 1, yet by our understanding, OTTR-seq must reach the end of its RNA template to "jump" to the second adapter, in order to be sequenced (unlike TruSeq or DM-tRNA-seq, which allows the second linker to be added after premature termination of cDNA synthesis). Thus, this may actually be analysis of slightly shortened RNA molecules due to RNA degradation or exonucleases, not premature cDNA extension. This distinction should be clarified in the manuscript for all methods discussed, as other methods such as NEBNext and ARM-seq also do not produce any sequencing reads for early-termination products (both adapters are ligated on to the RNA at the very beginning).

Overall, this is a convincing first study profiling an important new method that will clearly improve our ability to measure and understand tRNA dynamics and complexity in the cell.

We found quite a few technical issues that need to be addressed, particularly with the figures, but also with other missing details. We believe the results are sound, but in the manuscript's unpolished data presentation state, it was often difficult or frustrating to try to follow the analyses. We hope these suggestions are helpful in strengthening the clarity and overall impact of the study.

In general, figure axes need to be systematically fixed throughout. Often axes labels are missing or not explained, or the formatting makes them unreadable or overlapping the graphs.

Supplemental Table S1 appears to be missing, making it difficult to find the datasets produced in this study.

What are the accession numbers of the data analyzed from prior studies? Others will not be able to reproduce/verify these external data analyses without full details in the methods.

Failed replicate: For the data shown in Figure 2D (yeast, full length tRNA mismatch data), there are 2 biological WT replicates, but only 1 valid biological Trm1(-/-) replicate (we checked, the other replicate had almost no reads in the deposited data file and was essentially a failed library). The single replicate did support the observations in the paper that the Trm1(-/-) leads to loss of misincorporation at position 26, so no serious concern with the results, but there should be a note that the second replicate essentially failed (and that failed sample removed from GEO).

Question regarding size selections: Figure 1A shows a wide size range, even after gel-purification, which is attributed to RNA fragmentation. However, shouldn't the second gel purification at the cDNA stage (to remove dimers), remove the short fragments (and at this stage, it's DNA, so should not fragment). This suggests a potential issue with gel sizing. Perhaps the authors can clarify why so many short reads are still found after two rounds of sizing (the second at the cDNA stage).

Concern with definition of "full-length" tRNA reads and early termination: we suggest the cutoff for counting a full-length read is too strict. It appears that if the read doesn't extend ALL the way to position 1 it is deemed a "partial-length" read. For example, with how the OTTR enzyme works, some of these "partial-length" reads ending near the 5' end of the tRNA are probably bioinformatic artifacts (where the additional 5'nt in the OTTR reads were skipped and thus a trimming of the 1st base in the tRNA sequence). With how we understand OTTR works, it should NOT jump to the other adapter in the case of a stop due to a tRNA modification. It could be that these truncated tRNAs are in fact being degraded and the reads are reflective of those intermediate degradation products, instead of due to modifications causing pre-mature stops. This is a very important point to address carefully, as the community may be led to believe, incorrectly, that OTTR produces reads with early RT ends.

With this issue in mind, Figures 1B-1F are probably no longer justified or should be moved to the Supplement (after re-interpreting the results) because the cause of the shorter reads for OTTR-seq is ambiguous (purification issues, degradation during the library prep?). If new graphs like these are created here or elsewhere, axes need to be labeled.

Case in point: In Figures 2A and 2B, many of the strongest known "stop"-inducing modifications do NOT appear to cause drop-offs (early termination) of reads. This should have been a huge red flag that shorter OTTR-seq reads are NOT caused by early RT termination. To be clear, the RT reaction *may* be stopped, but you will NOT see those reads in the OTTR-seq output, because you will not have a second adapter added on to the end, required for PCR amplification.

In Figure 2C-2F, these plots actually aren't that helpful because there is way too much data shown. No axes labels in 2D-2F are large enough to read. Everything is packed in, thus making it too busy and hard to understand. The comparison of 2C to 2D is very difficult because they are different sizes. The color labeling of the tRNAs is not useful at this small scale. 2D could be shown in a box plot, showing lack of misincorporation at just this position across most tRNAs. The tRNA position these diamonds correspond to is not useful unless modifications are labeled and lined up in the same (legible) figure. 2F, which has the bulk of the modification information, is very very difficult to get useful information from, as it's unclear exactly what it's showing.

We highly recommend just using the corresponding tRAX output files which show the results much more cleanly and clearly – here is how much clearer this data could be displayed, just by using the default output files in tRAX:

https://www.dropbox.com/s/tlea5q66ng5jbu1/Figre2C-F-from-tRAX.pdf?dl=0

Fairness in tRF sequencing method comparisons: All of the direct abundance comparisons with other methods are ones that do NOT involve PNK or AlkB treatment prior to library generation. This is going to lead to VASTLY different tRF/tDR pools due to these factors alone. Thus, this is something of a straw-man comparison because more quantitative tDR sequencing methods have been in use (ARM-seq/PANDORA-seq) that would be more fair comparisons to the current state-of-the-art. We do not recommend removing this data or the comparisons, but we do believe that these differences in library prep (with "pre-2015" technology) should be clearly represented as weaker than other existing methods not tested. There was acknowledgement that newer methods like mim-tRNA-seq give roughly as good performance as OTTR-seq, but the same should be mentioned that newer tDR/tRF sequencing methods get much closer to OTTR-seq than those done in this study. That said, OTTR-seq gives you the best of both full-length and tDR/tRF sequencing in one go – which is really the quality that makes it outshine all other methods, from our perspective (which is not emphasized enough in our humble opinion).

While comparing across multiple types of other kinds of sequencing (experiments not carried out in this study), is it fair to make these comparisons when the data is also from multiple source types (Figure S2)? YAMAT and Hydro-seq are thrown into the same context of the other sequencing library preps under the umbrella of "extensive premature RT terminations", however, Hydro-tRNA-seq and ARM-seq can't produce sequencing reads with RT stops since they ligate 5' and 3' adapters prior to RT. YAMAT is given the appearance that it reads through tRNAs, but again, it only reads through the ones where the RT reaches the 5' adapter (thus you can't measure the true rate of RT stops because those stops, which do happen, are never observed in the seq data). The only library preps that could show RT-stops in this collection are QuantM and DM-tRNA-seq, from our understanding of the library prep protocols. Also, what exact data is being used here to assess these other methods? Some of these techniques have used both AlkB treated, and untreated. Did you use the AlkB+ samples in the datasets, or did you use -AlkB. This is never mentioned. All of these data sources and exactly which samples were used need to be specified clearly. These comparisons are extremely tricky, especially considering some of these methods were not designed to read full-length tRNAs, so doing direct comparisons and attributing differences to inefficient read-through of stops should be done with more care, separating the methods by their intended purposes.

Suggestions for improving Figure 3: Figure 3C could be presented more clearly with something like a stacked bar plot. It is hard to digest the changes with multiple bars for each isotype (just merge them unless a claim is made about a specific isotype/isodecoder). Additionally, it's not clear that the Y-axis metric being used (read count/1000) is the best for trying to compare these data across methods with such disparate scales. One can see that Gly has more reads in the NEB data than the others, but the other isotypes are very hard to compare at these scales (other than to see that they are different). 3D and 3E: Why is Fu_r1 shown as well as Fu_r2 (which look identical)?? And only Fu2018_r2 shown in 3C? (labeling inconsistency, BTW – 2018 included in one label but not the other). Figures 3D and 3E have axes that are impossible to read. Also, the Green/Red color coding for Starts and Stops seem to be backwards. RT-based sequencing reads "Start" at the 3' ends of tRNAs, and "End" towards the 5' end of the tRNA. These are labeled as being reads, so green should presumably be start, and red as stops. If instead you wish to present it in terms of tRNA coordinates (not the RT reaction which produces the read data), then the left most end would be green (start), and the right most would be red (stop) – this is counterintuitive because I (and most) know how cDNA reads are generated from RNA, but either way, Start as red and stop as green is really confusing.

Figure 3 and Supplemental Figure 4: Is this data using NEB with UMI? The NEB kit mentioned doesn't come with a UMI containing adapter, let alone two UMI containing adapters, as far as we know. Are these the adapters from the Fu 2018 protocol? If not, what are the adapters that are used? What size are the UMIs? This can greatly affect the amount of deduplication.

Suggestions for improving Figure 4: In Panel A, the different colored bars are nearly impossible to tell the difference between them. These would be better presented with another method. For Panel B, it is surprising so many ncRNAs are lumped into "other". For a useful comparison, it would be helpful to break out at least snoRNAs and miRNAs (and possibly snRNAs), as the recovery rate of these other small RNAs directly affect competition with tRNAs, and they are processed in the cell differently. Panel C is confusing because the X-axis is numbered but the colors are for the different isotypes. What are the numbers on the x-axis for? (this extends to figures S4 and S6). In Panel D, the naming convention for the tRNAs is inconsistent with the other figures (e.g. "Gly-GCC" in Figure 4 vs "tRNA-Gly-GCC" in other figures). Exactly how were the plots in Figure 4 created? Were UMI used? Without legible scales, we are just looking at shapes, which can be deceiving for very high or low abundance tRFs/tDRs.

Figure S4 panel B it also appears that the 5' and 3' colors in the legend are flipped (?) – aside from this, it is very hard to gain useful information from Figure S4B, perhaps it could be presented in a better way?

Data Analysis Issues: We downloaded the submitted GEO data, and noted that the naming conventions in the datasets are very confusing (yeast_17_miRvana), instead of giving clear info about the actual sample (and we can't seem to find any metadata for these, had to guess, making reproducing the analyses extremely difficult). Additionally, we find the description of the methods for read processing incomplete. Again, it is unclear how long the UMIs used were. Any special parameters for read processing (trimming extra bases in OTTR-seq reads) are also omitted, again making it very difficult for others to reproduce the results.

---

## [Author Response]

Essential Revisions:1) Better comparison of OTTR to existing methods, such as YAMAT-seq and Mim-tRNA seq. Primary data of Is the measurement of full length tRNA quantitative? The authors observe that the fraction of full length tRNAs measured compared to RT stops is high, like in YAMAT-seq and mim-tRNA-seq. One possibility that the authors detect a high fraction of full length tRNAs is because cDNA products resulting from RT stops are systematically not detected, perhaps due to template switching being prevented by the cDNA template still annealed to the tRNA. YAMAT-seq is notorious in overemphasizing full-length tRNA as only the full-length cDNA product of tRNA is PCR-amplified and sequenced. Mim-tRNA-seq on the other hand, showed that the entire cDNA products BEFORE PCR amplification are indeed mostly full-length. The authors need to show primary data of the cDNA libraries before PCR amplification to justify this claim. In addition, the authors should show without PCR amplification that the OTTR RT is indeed superior in reading through e.g. yeast tRNAPhe yW37 modification (reported in 1C) compared to TGIRT and the thermostable superscript IV used in other recent tRNA-seq protocols.

See Introduction. We now compare OTTR systematically to all existing methods, presented visually in Figures S2-3, and described much more extensively in the Discussion. The only aspect of this comment that we have not addressed is the specific request to demonstrate superiority of OTTR RT in reading through modifications like yW37. The central issue with addressing this comment experimentally is that OTTR is a low input protocol – using higher inputs of starting material leads to biased capture of specific RNAs; keeping input levels roughly equimolar with primer concentration is essential for unbiased library construction – and so we do not have sufficient cDNA to visualize without amplification (we carry out size selection for libraries “blind,” based only on size markers). In Figure 5 we do show that despite its efficient capture of intact tRNAs even in untreated conditions, removing certain nucleotide modifications (via AlkB demethylation for relevant substrates, or genetically for others) improves capture of some tRNA species, highlighting that even for the OTTR enzyme, nucleotide modifications present a barrier to reverse transcription.

2) Demonstrate better that OTTR actually distinguishes between 3' tRNA fragment and RT stops. Sequencing reads mapping only to the 3' half are interpreted as fragments.This is confusing because the method implied that only full length products can be amplified in libraries. The cause and confirmation of the 3' fragments need to be better substantiated.

The reviewer may be confusing the data for full length tRNA libraries (Figures1-2) and small RNA libraries (Figures3-4) in this paper. In the context of *full length* tRNA libraries we do not consider short products to be tRNA fragments: these are almost certainly premature RT termination products. Focusing on the 3’ reads interpreted as 3’ fragments: we have very clear evidence regarding the origin of these sequencing reads. For 3’ halves in *small RNA libraries* – which are the only context in which we interpret 3’ reads as “fragments” – the data in Figure S4A-B show a very clear distinction between the 3’ reads obtained before and after RNY1 expression. The former represent a mixture of tRNA degradation products presumably arising from RNA isolation and handling, along with any premature RT stops (which are almost certainly quite rare in these libraries; we find very few such 30-40 nt reads in libraries prepared from intact tRNAs). Importantly, these 3’ reads exhibit extensive heterogeneity in 5’ ends. Conversely, fragments isolated following RNY1 overexpression – where we confirm by Northern blotting that tRNAs are indeed extensively cleaved – exhibit precise 5’ ends, and as quantified using spike ins are at least 10X more abundant than the 3’ reads in no RNY1 conditions. We therefore argue that 3’ fragments are easily distinguishable from premature RT stops in OTTR datasets.

Reviewer #1 (Recommendations for the authors):Gustafsson et al. describe an application of the recently developed OTTR library prep method to tRNA and tRNA fragments. They present characterization of tRNA-sequencing results from yeast and mouse testis/spermatozoa. The authors claim a significant improvement in the detection of full length tRNAs compared to other techniques; importantly, the authors further focus on the detection of tRNA fragments which are biologically important, but systematically under-detected by existing techniques. The authors show that OTTR detects select tRNA fragments better than a couple of commercial kits. This reviewer shares the author's enthusiasm about the importance of detecting tRNA fragments, especially 3' tRNA fragments, and further share their enthusiasm about the potential for OTTR. However, it is unclear that using OTTR towards the study of tRNA and tRNA fragments is a substantive advance over the several protocols published in the last 3 years. If the authors can more substantively demonstrate how OTTR compares to other methods quantitatively and include additional controls for sequencing results, this could be a method of great utility.1. Figure 1: Is the measurement of full length tRNA quantitative? The authors observe that the fraction of full length tRNAs measured compared to RT stops is high, like in YAMAT-seq and mim-tRNA-seq. One possibility that the authors detect a high fraction of full length tRNAs is because cDNA products resulting from RT stops are systematically not detected, perhaps due to template switching being prevented by the cDNA template still annealed to the tRNA. YAMAT-seq is notorious in overemphasizing full-length tRNA as only the full-length cDNA product of tRNA is PCR-amplified and sequenced. Mim-tRNA-seq on the other hand, showed that the entire cDNA products BEFORE PCR amplification are indeed mostly full-length. The authors need to show primary data of the cDNA libraries before PCR amplification to justify this claim. In addition, the authors should show without PCR amplification that the OTTR RT is indeed superior in reading through e.g. yeast tRNAPhe yW37 modification (reported in 1C) compared to TGIRT and the thermostable superscript IV used in other recent tRNA-seq protocols.

This issue is broadly addressed in Figures S2-3, where we compare OTTR to a large number of published tRNA-focused sequencing protocols. The specific issue in RT readthrough of yW37 etc. is addressed in our response to Essential Revision 1, above.

2. Figure 1: The relative quantification of individual tRNAs needs to be done for reporting any new tRNA-seq method. At the least the authors should compare the yeast OTTR results with that from the mim-tRNA-seq paper.

Done as requested. Addressed in Figures S2-3.

3. Figure 2: 2A and B show that Lys-CTT has starts near the anticodon loop at t6A, indicating fragments. But this effect is eliminated by trm1 deletion. An alternative from 5' fragments is that the RT reinitiates downstream of a stop induced by a m22G or t6A modification.

I am not sure what is requested here. Our original contention was that these are premature RT stops (not RT reinitiation), not biological fragments, which I believe is roughly consistent with what the reviewer believes as well?

4. RNA and read lengths appear inconsistent. First, from figure 1A, is miRvana the best source for capturing small tRNA fragments? More 30-nt material is retained with gel purification. Further, the insert sizes between figures 1A, 3B and 4A are confusing. 1A suggests majority full length tRNA, while figures 3B and 4A crop out this size of read, preventing comparison of relative levels of fragment and full length tRNA. Additionally, figure 1C suggests that Serine and leucine tRNAs are not captured faithful as full length. This could be due to these tRNAs being type II (~90nt long) but sequenced with only 75bp Illumina reads.

Regarding the miRvana question: we find that, surprisingly, gel purification of full length tRNAs prior to cloning results in more tRNA fragmentation than does simple miRvana size selection. Although counterintuitive – why should more stringent size selection lead to more variable RNA lengths? – this has been seen by other investigators as well (via personal communication – not sure I can find a citation), and our empirical evidence is very clear on this point.

Regarding the insert sizes in Figures 1A vs 3B and 4A: Figures1-2 focus on full length tRNA sequencing libraries, while Figures3-4 focus on small RNA sequencing libraries. The size distributions reflect the intended goal of the dataset in question.

Finally, the serine and leucine comment is spot on – in new Figure S1B we show that serine and leucine tRNAs are underrepresented as a result of our use of 75 bp Illumina sequencing reads, which fail to read all the way through these longer tRNAs. This was an unfortunate oversight on our part, and is now noted in the revised manuscript.

5. Can OTTR actually distinguish between 3' tRNA fragment and RT stops? Figure 1B is the strongest evidence to distinguish, and my reading is that reads mapping only to the 3' half must be fragments since this method doesn't suffer from RT stops. This reasoning is insufficient to interpret 3' fragments in unknown biological systems, which severely limits the utility of this exciting approach.

See response to Essential Revision 2, above.

6. The biological advances of this work are unclear. The primary biological discovery presented is that the suite of small RNAs in mouse testis/sperm is different from previous measurements. Even for this result no Northern blot validation is provided for the OTTR data.

The presence of 3’ tRNA fragments in sperm was previously shown by Northern blotting by our group and the Chen group in 2018. The sncRNA payload of mature sperm is of extreme interest to a burgeoning paternal effect community and in our opinion is an extremely important insight. It will certainly be highly cited, anyway.

Reviewer #2 (Recommendations for the authors):The experimental design for this study was effective in choice of samples and treatments compared. Use of both yeast and mammalian samples that had been previously analyzed was a wise choice by which to observe tRNA pool complexity and misincorporations caused by modifications. We also appreciate the use of tRAX software, which specifically deals with the complexity of misincorporations caused by tRNA modifications, tRNA mapping ambiguity, and differential expression (full disclosure: developed in the Lowe lab). The use of trm1 mutants, inducible RNY1 overexpression, and PNK treatment were all effective experimental contexts showing the relative performance of the various sequencing methods tested. Some readers may feel the study was a bit limited in scope in that it did not do direct sample comparisons with the other leading methods for analyzing full length tRNAs (DM-tRNA-seq, mim-tRNA-seq and/or YAMAT-seq), relying instead on data previously published by other groups. However, in the authors' prior studies, they focused heavily on tRNA fragments in the mouse germline and utilized traditional small RNA-seq methods like TruSeq, which yielded very high-profile results based on high abundance of 5' halves of only a handful of tRNAs. This new study successfully argues the importance of re-examining those prior results given the effectiveness of this new generation of sequencing methods. OTTR-seq is arguably at the front of these newer methods, given its flexibility to measure both tRNAs and tRNA-derived small RNAs at the same time, both in terms of abundance and the misincorporations caused by Watson-Crick face interfering modifications. If all future tRNA-seq studies used OTTR-seq (or equally well performing methods), the tRNA field would take a major step forward.Aside from easily addressed issues with figures, the only serious confusion that needs to be addressed is the assessment or interpretation of "early termination" sequencing reads. An analysis of premature termination was shown in Figure 1, yet by our understanding, OTTR-seq must reach the end of its RNA template to "jump" to the second adapter, in order to be sequenced (unlike TruSeq or DM-tRNA-seq, which allows the second linker to be added after premature termination of cDNA synthesis). Thus, this may actually be analysis of slightly shortened RNA molecules due to RNA degradation or exonucleases, not premature cDNA extension. This distinction should be clarified in the manuscript for all methods discussed, as other methods such as NEBNext and ARM-seq also do not produce any sequencing reads for early-termination products (both adapters are ligated on to the RNA at the very beginning).Overall, this is a convincing first study profiling an important new method that will clearly improve our ability to measure and understand tRNA dynamics and complexity in the cell.We found quite a few technical issues that need to be addressed, particularly with the figures, but also with other missing details. We believe the results are sound, but in the manuscript's unpolished data presentation state, it was often difficult or frustrating to try to follow the analyses. We hope these suggestions are helpful in strengthening the clarity and overall impact of the study.In general, figure axes need to be systematically fixed throughout. Often axes labels are missing or not explained, or the formatting makes them unreadable or overlapping the graphs.

We have attempted to fix these issues throughout.

Supplemental Table S1 appears to be missing, making it difficult to find the datasets produced in this study.

Fixed.

What are the accession numbers of the data analyzed from prior studies? Others will not be able to reproduce/verify these external data analyses without full details in the methods.

Added as requested (Comparison to published datasets, Methods).

Failed replicate: For the data shown in Figure 2D (yeast, full length tRNA mismatch data), there are 2 biological WT replicates, but only 1 valid biological Trm1(-/-) replicate (we checked, the other replicate had almost no reads in the deposited data file and was essentially a failed library). The single replicate did support the observations in the paper that the Trm1(-/-) leads to loss of misincorporation at position 26, so no serious concern with the results, but there should be a note that the second replicate essentially failed (and that failed sample removed from GEO).

Altered as requested.

Question regarding size selections: Figure 1A shows a wide size range, even after gel-purification, which is attributed to RNA fragmentation. However, shouldn't the second gel purification at the cDNA stage (to remove dimers), remove the short fragments (and at this stage, it's DNA, so should not fragment). This suggests a potential issue with gel sizing. Perhaps the authors can clarify why so many short reads are still found after two rounds of sizing (the second at the cDNA stage).

We do not exactly understand why shorter reads persist through the double size selection. Our only hypothesis is that the persistence of short molecules through the second size selection step arises from the distribution of molecules of a given size during gel electrophoresis – eg if one were to run a pure 100 bp DNA on a gel and cut out the gel corresponding to 70-80 bp DNA, some subset of the 100 bp molecules would nonetheless be isolated as DNA migration through a gel results in a band which represents the center of a distribution. As far as the issue with double size selection, it is very clear in our data that a first size selection step counterintuitively leads to increased RNA fragmentation, and this has been informally confirmed by many of our colleagues. We feel the point is important to make here.

Concern with definition of "full-length" tRNA reads and early termination: we suggest the cutoff for counting a full-length read is too strict. It appears that if the read doesn't extend ALL the way to position 1 it is deemed a "partial-length" read. For example, with how the OTTR enzyme works, some of these "partial-length" reads ending near the 5' end of the tRNA are probably bioinformatic artifacts (where the additional 5'nt in the OTTR reads were skipped and thus a trimming of the 1st base in the tRNA sequence). With how we understand OTTR works, it should NOT jump to the other adapter in the case of a stop due to a tRNA modification. It could be that these truncated tRNAs are in fact being degraded and the reads are reflective of those intermediate degradation products, instead of due to modifications causing pre-mature stops. This is a very important point to address carefully, as the community may be led to believe, incorrectly, that OTTR produces reads with early RT ends.

This is related to the concern in Essential Revision 1, and addressed there: we would love to be able to show a gel of the cDNA from the first RT step, but the input levels for OTTR are so low that the cDNA in these gels is invisible.

With this issue in mind, Figures 1B-1F are probably no longer justified or should be moved to the Supplement (after re-interpreting the results) because the cause of the shorter reads for OTTR-seq is ambiguous (purification issues, degradation during the library prep?). If new graphs like these are created here or elsewhere, axes need to be labeled.Case in point: In Figures 2A and 2B, many of the strongest known "stop"-inducing modifications do NOT appear to cause drop-offs (early termination) of reads. This should have been a huge red flag that shorter OTTR-seq reads are NOT caused by early RT termination. To be clear, the RT reaction *may* be stopped, but you will NOT see those reads in the OTTR-seq output, because you will not have a second adapter added on to the end, required for PCR amplification.

We would be willing to move these figures to the Supplement but it is our strong preference that they remain, with the new suitably modified text more extensively describing potential interpretations.

In Figure 2C-2F, these plots actually aren't that helpful because there is way too much data shown. No axes labels in 2D-2F are large enough to read. Everything is packed in, thus making it too busy and hard to understand. The comparison of 2C to 2D is very difficult because they are different sizes. The color labeling of the tRNAs is not useful at this small scale. 2D could be shown in a box plot, showing lack of misincorporation at just this position across most tRNAs. The tRNA position these diamonds correspond to is not useful unless modifications are labeled and lined up in the same (legible) figure. 2F, which has the bulk of the modification information, is very very difficult to get useful information from, as it's unclear exactly what it's showing.

We respectfully disagree with these comments. Highlighting aspects of genome-wide datasets typically includes showing individual examples of sequencing reads or of specific species (as in Figure 2A), and then showing the entire dataset in various summary figure panels (as in Figures 2C-D), and specific cross sections of the dataset (as in Figure 2F). We are unsure why the reviewer thinks that “it’s unclear exactly what [Figure 2F is] showing” – the figure legend makes it very clear? We actually feel Figure 2F precisely addresses the reviewer’s concern about Figures 2C-D being too dense, so this overall comment strikes us as a little bit self-contradictory?

We highly recommend just using the corresponding tRAX output files which show the results much more cleanly and clearly – here is how much clearer this data could be displayed, just by using the default output files in tRAX:https://www.dropbox.com/s/tlea5q66ng5jbu1/Figre2C-F-from-tRAX.pdf?dl=0

We respectfully disagree that standard tRAX outputs are clearer – we find the dots too small, and the graph paper lines everywhere on some of the plots are distracting and make the data difficult to see. Our choices of visual presentation are more consistent with our preferences in terms of being able to visually absorb data.

Fairness in tRF sequencing method comparisons: All of the direct abundance comparisons with other methods are ones that do NOT involve PNK or AlkB treatment prior to library generation. This is going to lead to VASTLY different tRF/tDR pools due to these factors alone. Thus, this is something of a straw-man comparison because more quantitative tDR sequencing methods have been in use (ARM-seq/PANDORA-seq) that would be more fair comparisons to the current state-of-the-art. We do not recommend removing this data or the comparisons, but we do believe that these differences in library prep (with "pre-2015" technology) should be clearly represented as weaker than other existing methods not tested. There was acknowledgement that newer methods like mim-tRNA-seq give roughly as good performance as OTTR-seq, but the same should be mentioned that newer tDR/tRF sequencing methods get much closer to OTTR-seq than those done in this study. That said, OTTR-seq gives you the best of both full-length and tDR/tRF sequencing in one go – which is really the quality that makes it outshine all other methods, from our perspective (which is not emphasized enough in our humble opinion).

This is now addressed in Figures S2-3 and revised Results and Discussion.

While comparing across multiple types of other kinds of sequencing (experiments not carried out in this study), is it fair to make these comparisons when the data is also from multiple source types (Figure S2)? YAMAT and Hydro-seq are thrown into the same context of the other sequencing library preps under the umbrella of "extensive premature RT terminations", however, Hydro-tRNA-seq and ARM-seq can't produce sequencing reads with RT stops since they ligate 5' and 3' adapters prior to RT. YAMAT is given the appearance that it reads through tRNAs, but again, it only reads through the ones where the RT reaches the 5' adapter (thus you can't measure the true rate of RT stops because those stops, which do happen, are never observed in the seq data). The only library preps that could show RT-stops in this collection are QuantM and DM-tRNA-seq, from our understanding of the library prep protocols. Also, what exact data is being used here to assess these other methods? Some of these techniques have used both AlkB treated, and untreated. Did you use the AlkB+ samples in the datasets, or did you use -AlkB. This is never mentioned. All of these data sources and exactly which samples were used need to be specified clearly. These comparisons are extremely tricky, especially considering some of these methods were not designed to read full-length tRNAs, so doing direct comparisons and attributing differences to inefficient read-through of stops should be done with more care, separating the methods by their intended purposes.

Addressed in Figures S2-3 and revised Results and Discussion.

Suggestions for improving Figure 3: Figure 3C could be presented more clearly with something like a stacked bar plot. It is hard to digest the changes with multiple bars for each isotype (just merge them unless a claim is made about a specific isotype/isodecoder). Additionally, it's not clear that the Y-axis metric being used (read count/1000) is the best for trying to compare these data across methods with such disparate scales. One can see that Gly has more reads in the NEB data than the others, but the other isotypes are very hard to compare at these scales (other than to see that they are different). 3D and 3E: Why is Fu_r1 shown as well as Fu_r2 (which look identical)?? And only Fu2018_r2 shown in 3C? (labeling inconsistency, BTW – 2018 included in one label but not the other). Figures 3D and 3E have axes that are impossible to read. Also, the Green/Red color coding for Starts and Stops seem to be backwards. RT-based sequencing reads "Start" at the 3' ends of tRNAs, and "End" towards the 5' end of the tRNA. These are labeled as being reads, so green should presumably be start, and red as stops. If instead you wish to present it in terms of tRNA coordinates (not the RT reaction which produces the read data), then the left most end would be green (start), and the right most would be red (stop) – this is counterintuitive because I (and most) know how cDNA reads are generated from RNA, but either way, Start as red and stop as green is really confusing.

We have removed the redundant Fu_2018 libraries from Figure 3D-E. As far as the axes on these figures, the goal of the figure panel is to show the “shape” of sequencing reads across the tRNA – the y axis could almost be considered arbitrary units in terms of the desired information transfer here. We do not think adding 6 point font “1000”s and so forth all over these graphs would improve the legibility of the figure. As for the red and green, the reviewer makes a reasonable point but correctly highlights the issue – we are using “start” and “stop” relative to the tRNA coordinate, while sequencing read starts and stops are inverted. As far as the red and green they were not actually chosen based on traffic signals, we apologize for this confusing choice – if the editors find this important we can change the color scheme but we feel it is a minor issue requiring substantial work to deal with and so have left the colors intact in this revised manuscript.

Figure 3 and Supplemental Figure 4: Is this data using NEB with UMI? The NEB kit mentioned doesn't come with a UMI containing adapter, let alone two UMI containing adapters, as far as we know. Are these the adapters from the Fu 2018 protocol? If not, what are the adapters that are used? What size are the UMIs? This can greatly affect the amount of deduplication.

Addressed in the revised Methods.

Suggestions for improving Figure 4: In Panel A, the different colored bars are nearly impossible to tell the difference between them. These would be better presented with another method. For Panel B, it is surprising so many ncRNAs are lumped into "other". For a useful comparison, it would be helpful to break out at least snoRNAs and miRNAs (and possibly snRNAs), as the recovery rate of these other small RNAs directly affect competition with tRNAs, and they are processed in the cell differently. Panel C is confusing because the X-axis is numbered but the colors are for the different isotypes. What are the numbers on the x-axis for? (this extends to figures S4 and S6). In Panel D, the naming convention for the tRNAs is inconsistent with the other figures (e.g. "Gly-GCC" in Figure 4 vs "tRNA-Gly-GCC" in other figures). Exactly how were the plots in Figure 4 created? Were UMI used? Without legible scales, we are just looking at shapes, which can be deceiving for very high or low abundance tRFs/tDRs.

For Figure 4B, the goal is to focus on the tRNA and rRNA fragments, which increase in abundance following PNK treatment to resolve cyclic 2’-3’ phosphates produced by RNaseA/T family enzymes. snoRNAs and miRNAs in sperm are not subject to such cleavage. We have removed the x axis numbers in Figure 4C as requested. We have fixed the tRNA labeling in panel D as requested. We have clarified the data used throughout this figure, as requested. Finally, if the editors wish we can increase the font size on the panel D y axes but as mentioned above we think having lots of six point numbers showing on the number of panels here would be distracting, and we actually think the goal here is precisely to “just look at shapes”. For the reviewer’s information however these are among the more abundant tRFs in the Truseq and NEB datasets, as they exhibit such limited complexity (eg Figure 4C) that most other species are indeed very low abundance and extremely noisy-looking.

Figure S4 panel B it also appears that the 5' and 3' colors in the legend are flipped (?) – aside from this, it is very hard to gain useful information from Figure S4B, perhaps it could be presented in a better way?

Unclear why the reviewer thinks the colors are flipped here? We agree that this (now Figure S4D) is not easy to extract specific data points from, but the purpose of the figure is to show differences in complexity (eg sparse bars in the top panel compared to a forest of bars for OTTR) and bias towards specific tRNA ends (bars of both colors in OTTR). Which we believe is indeed communicated in the figure as it stands. Also, we cannot think of a better visualization for this point. The figure panel could be deleted but we feel it does add a tiny bit to the manuscript so prefer to keep it.

Data Analysis Issues: We downloaded the submitted GEO data, and noted that the naming conventions in the datasets are very confusing (yeast_17_miRvana), instead of giving clear info about the actual sample (and we can't seem to find any metadata for these, had to guess, making reproducing the analyses extremely difficult). Additionally, we find the description of the methods for read processing incomplete. Again, it is unclear how long the UMIs used were. Any special parameters for read processing (trimming extra bases in OTTR-seq reads) are also omitted, again making it very difficult for others to reproduce the results.

We have updated the metadata as requested, as well as attempting to make the dataset titles more intuitive.